# A simple biota removal algorithm for 35-GHz cloud radar measurements

Madhu Chandra R. Kalapureddy[1*], Patra Sukanya[1,3], Subrata K Das[1], Sachin M Deshpande[1], Govindan Pandithurai[1], Andrew L Pazamany[2], Jha K Ambuj[1], Kaustav Chakravarty[1], Prasad Kalekar[1], Hari Krishna Devisetty[1] and Sreenivas Annam[4]

[1]Indian Institute of Tropical Meteorology (IITM), Dr Homi Bhabha road, Pashan, Pune 411008, Maharastra, India.
[2]Prosensing Inc., 107 Sunderland road, Amherst, MA 01002, US.
[3]Savitribai Phule Pune University, Pune 411007, India.
[4]VFSTR University, Vadlamudi 522213, India.

*Correspondence to:* M.C.R. Kalapureddy (kalapureddy1@gmail.com)

**Abstract.** Cloud radar reflectivity profiles can be an important measurement for the investigation of cloud vertical structure in a resourceful way. However, extracting intended meteorological cloud content from the overall measurement often demands an effective technique or algorithm that can reduce error and observational uncertainties in the recorded data. In this work, a technique is proposed to identify and separate cloud and non-hydrometeor echoes using the radar Doppler spectral moments profile measurements. The point and volume radar target based theoretical radar sensitivity curves are used for remove the receiver noise floor and identified radar echoes scrutinized for the signal de-correlation period. It is hypothesizing that cloud echoes are observed to be temporally more coherent, homogenous and have a longer correlation period than biota. That can be checked statistically using ~4 second sliding averaged reflectivity profiles mean and standard deviation value. The above step critically helps in screen out clouds by filtering biota. The final important step strives for the retrieval of cloud height. The proposed algorithm potentially identify cloud height solely through the systematic characterization of Z variability using the local atmospheric vertical structure knowledge besides to the theoretical, statistical, and echo tracing tools are the key components with this study. Thus, characterization of high resolution cloud radar reflectivity profile measurements with the theoretical echo sensitivity curves and observed echo statistics for the cloud height tracking (TEST). TEST show superior performance in screen out cloud and filtering out isolated insects. TEST constrained with polarimetric measurements found more promise under high density biota whereas TEST combined LDR and Spectral Width (SW) perform potentially to filter out biota within high turbulent cumulus clouds. This TEST technique is promisingly simple in realization but powerful in performance due to constraining flexibility for identifying and filtering out the biota and screen out the true shallow cumulus clouds in the convective boundary layer (CBL). The low level CBL cumulus clouds are strongly linked to the rain making mechanism associated with the cloud vertical structure (CVS). CVS associated with Indian Summer Monsoon (ISM) region hold a key factor in improve the ISM tropical cloud characterization and hence the predictability of cloud feedback in a changing climate.

## 1.0    Introduction

Short wavelength (millimeter-wave) Doppler radars are well known as cloud radars for their high sensitivity that is required to sense the cloud droplets or ice crystals to infer cloud properties at high resolution (e.g., Lhermitte, 1987; Pazmany et al., 1994; Frisch et al., 1995; Kollias and Albrecht, 2000; Sassen et al., 1999; Hogan et al., 2005). The atmospheric radar echoes in the optically clear boundary layer are mainly either from Bragg scattering through refractive index irregularities due to turbulence in the atmosphere (wind profilers; e.g., Ecklund et al., 1988; Gossard 1990)  or particle scattering from hydrometeors and biota which is air-borne biological targets such as birds and insects, and waste plant materials e.g., dry leaves, pollen or dust (also known as "atmospheric plankton" or atmospheric "biota" or simply "insects"; Wilson et al., 1994; Lhermitte, 1966; Clothiaux et al., 2000; Teschke et al., 2006;). Although insects (hereafter biota) are probably the principal contaminants because of their size and dielectric constant, spiders, spider webs, and other organic materials have been detected in the atmosphere through the use of nets and other means (Sekelsky et al., 1998). Furthermore due to reduced scattering efficiency in the Mie region, cloud radar observations at 95 GHz are found to be less (~5 dBZ) sensitive to biota than observations at 35 GHz (Khandwalla et al., 2003).  Cloud radar signals frequently encounter this biota, within a couple of kilometers altitude close to the Earth surface, confined mostly to the Atmospheric Boundary Layer (ABL). These echoes from the biota in the ABL have reflectivity values comparable to those from the clouds, and thus they contaminate and mask the true cloud returns (Luke et al., 2008). Though the nature of shallow clear air radar echoes was first doubtful, but later, these echoes over land in the CBL were proved to be contaminated by particle scattering from biota rather than to refractive index gradients (e.g., Gassard 1990; Russell and Wilson, 1997). Importantly the nature of clear-air echoes are a nuisance for radar based studies on CBL clouds since they may contaminate the true cloud echo (e.g., Martner and Moran, 2001). However, these clear-air echoes can be advantageous in understanding and characterizing the CBL (e.g., Chandra et al., 2010; 2013). But in order to utilise the potential purpose of cloud radar for studying clouds, one needs to identify and preserve the true cloud echoes from biota contamination that is mostly confined within the atmospheric boundary layer (ABL). The ABL shallow/ low level cumulus clouds are strongly linked to the rain making mechanism at lower region of the cloud vertical structure and hold a  key factor in predictability of cloud feedback in a changing climate (Tiedtke 1989; Bony et al.2006; Teixeira et al. 2008) but their representation remain unresolved in large scale modeling. This gives rises to the need of most possible unbiased and systematic observational study of shallow cumulus cloud to unravel its morphological as well as characteristic features. Therefore, the current work focuses on identifying and filtering biota echoes in order to significantly improve the quality of cloud radar data. This allows better characterization of the tropical Cloud Vertical Structure.

Review of previous studies shows that different techniques have been attempted to remove non hydrometeor echoes, for example, static techniques for the ground clutter (Harrison et al., 2014; 2000), return signal-level correction (Doviak and Zrni´c, 1984; Torres and Zrni´c, 1999; Nguyen et al., 2008), dynamic filtering (Steiner and Smith, 2002), and operational filtering (Alberoni et al., 2003; Meischner et al., 1997). The aforementioned studies were mostly confined to the use of single polarization radar. However, a new possibility has been developed

using dual-polarization information to identify the non-meteorological clutter echoes (Zrnic´ and Ryzhkov, 1998;
Mueller, 1983; Zhang et al., 2005). With the advent in Doppler spectral processing, it is possible to have improved
clutter mask (Bauer-Pfundstein and Görsdorf, 2007; Luke et al., 2008; Warde and Torres, 2009; Unal, 2009). As
mentioned, one of the non-hydrometeor echoes is due to the insects and air-borne biota and these unwanted echoes
are problematic for studies involving meteorological information such as wind measurements (Muller and Larkin,
1985) and true cloud returns (Martner and Moran, 2001). As a consequence, observations of biota were done using
variable polarization and multiple frequency radars operating initially in the centimeter wavelength (Hajovsky et al.,
1966; Hardy et al., 1966; Mueller and Larkin, 1985). At millimeter wavelength radar, Bauer-Pfundstein and
Görsdorf (2007) showed effective LDR filtering of biota while Khandwalla et al. (2003) and Luke et al. (2008)
showed that dual-wavelength ratio filters are more effective than the linear depolarization ratio filters. Dual-
polarization also offers a wide variety of methods (e.g., Gourley et al., 2007; Hurtado and Nehorai, 2008; Unal,
2009; Chandrasekar et al., 2013). Fuzzy logic classification techniques for the identification and removal of spurious
echoes from radar are also in use (e.g., Cho et al., 2006; Dufton and Collier, 2015; Chandra et al., 2013). From the
above summary, it is therefore evident that most of the studies either concentrate on the polarimetric capabilities of
radar or computationally intensive spectral processing of radar data to filter out echoes contaminated by non-
hydrometeor targets. The importance of the current work presented here lies in the development of an algorithm that
uses solely high spatial and temporal resolution reflectivity measurements. These high spatial and temporal
resolution (25 m and 1 sec) measurements enable the characterization of irregular echoes associated with the
spurious nature of radar returns due to biota. This method is simple and does not require spacious complex spectral
data (and associated complicated analysis) or expensive advanced dual-polarimetric or dual-wavelength techniques.

**2.0    System, Data and Methodology**
This investigation employs vertically oriented Doppler spectral moments profile observations of IITM's
Ka-band scanning polarimetric radar (KaSPR) for the study of vertical cloud structure. In details, KaSPR employs
an improved variation of the well known Linear Frequency Modulated (LFM) pulse compression technique. The
KaSPR pulse compression technique is amplitude taper (window) (using a Tukey taper with 0.7 taper coefficient;
Window function) on the transmitted LFM pulse and the compression is implemented in the digital signal processor
system using a least mean squared filter (Mudukutore et al., 1998) to achieve much improved (lower) range side
lobes, compared to un-tapered LFM pulse compressed with a matched filter. Thus, KaSPR uses the 3.3 µs pulse
length with 10X LFM chirp compression with effective range resolution of 50 m (i.e., compressed to 0.33 µs) and
sampling in range (range gate spacing) at every 25 m with pulse reception frequency of 5 kHz. So, the radar data set
used for this work has the range samples at every 25 m with start range gate available are at 942 m AGL.  KaSPR
has been providing high resolution (25 m and 1 sec.) resourceful measurements of cloud and precipitation at a
tropical site (Mandhardev, $18.0429^0$ N $73.8689^0$ E, 1.3 km AMSL) on a mobile platform since June, 2013. Its other
main technical features are given in Table 1. KaSPR possesses sensitivity of ~ -60 (-45) dBZ at 1(5) km, it is
therefore sensitive to the cloud droplet. According to T-matrix Rayleigh computations, single 0.1 mm size of target
at ~35 GHz may have the reflectivity ~ -60 dBZ whereas, near 63 (1000000) of 0.05 (0.01) mm size is required to
give the same reflectivity. Furthermore in one second if there are 5000 (pulses per second) hits on the target in the
radar scattering volume, the mean of those 5000 samples at a range bin (height) will be affected by the mean
characteristics of target such as composition, orientation, number density and kinematics associated with it.
Therefore, it is safer to assume that the atmospheric or meteorological targets (in this case cloud particle) are
distributive in nature and passive in the sense that their motion and/or orientation are in resonance with the
kinematics of the background atmosphere. By comparison birds and insects are point targets in nature and active in
the sense that they can change their motion, direction and orientation within a few seconds. This leads to the
irregular nature of intermittent or spurious radar returns characteristic of atmospheric biota due to the much smaller
de-correlation time associated with them. This study utilizes the high resolution profile of cloud radar reflectivity
factor (Z) to construct the cloud vertical structures by filtering out the returns from the noise and biota.
Figure 1a represents the height profiles of $0^{th}$ moment (radar echo peak power) based Z on 27 Apr 2014 at
2303 UT with various theoretical radar sensitivity (noise-equivalent reflectivity, NER) curves (S0-S5; the range
profile correction with the start range sensitivity value of reflectivity, i.e., $r^2 x Z_{start\ range}$, where r is range or height and
Z is reflectivity, for S1, Z is -60 dBZ,  for example). These different NER or sensitivity curves are utilized to qualify
the observed radar returns that are indeed above the NER, the inherent radar receiver noise level. The receiver noise
level is the inherent thermal noise associated with electronic components in the receiver chain and also of other
sources which are taken into account through the noise figure (Table 1) and it remains approximately constant over
the length of the pulse returns. However, range correction is intuitive in the radar equation due to the decrease in
echo signal strength with increasing height (for vertical orientation). In order to determine the noise range in every
range bin, S0 to S5 are computed and overlaid on Z.  This allows for identification and characterization of the signal
that overlays the background system noise level. As discussed earlier, the signal at any level may have contributions
due to either volumetric meteorological cloud particulates and/or strong non-meteorological/non-hydrometeor point
targets (e.g. biota). In Figure 1a the echoes at ~3.7 km and below 2 km can be marked as cloud and biota
respectively as it exceeds the profile S5. The noise variations around 15 dB are mostly confined in between S0 and
S2 with S1 as mean NER.  Contrasting echo texture associated with the cloud and atmospheric biota is evident from
the height-time-intensity (HTI) plot of Z in Figure 1b.  This is a weak cloud case having reflectivity ~ -38 dBZ at
~3.7 km altitude with the presence of intermittent, non homogeneous echo texture from the biota below 2.7 km
altitude. Near similar weak cloud case of -38±2 dBZ at 5.4 km altitude is confirmed as cloud with the sharp increase
in relative humidity of ~ 80% at that altitude by collocated GPS-RS measurements but is not shown here (see Figure
A2). Biota echoes are observed to be confined most densely below 1.7 km and fall in the reflectivity range of -50 to
-20 dBZ. The observed standard deviation (S.D) is always more than 2 dBZ and in directly inferring de-correlation
period of ~4-5 sec (returns due to biota are observed to vanish at an interval of ~3-8 sec; see the lower part of the
HTI plot). On the de-correlation period, it is hypothesizing here that the running mean and standard deviation of ~4
seconds sliding window reflectivity profiles work in identifying all non-hydrometeor returns. Furthermore, the time
coherence of radar returns at every range sample can be checked for every 4 seconds as window period to infer the
echo power de-correlation time or degree of coherence period associated with biota return based on the S.D of Z
value. Two sensitivity (S1 and S5) tests have been performed on Z profile to quantify as noise floor, biota and the
meteorological cloud returns. All the tests have been affected due to the presence of non-meteorological echo due to
biota even though these are mostly present in the ABL.  Reflectivity values associated with the cloud boundaries are
very faint and are noticed to be fall within or close to system noise floor by 2-5 dBZ. The profile S5 seems to be
better in screening out the cloud echoes by 10 dBZ higher level than system mean noise floor but this can eliminate
significant portion of the weakest reflectivity area at the cloud edge (Figure 1d). Apart from clouds, biota also shows
higher reflectivity values than S5. Figure 1d is similar to Figure-1b except, it is completely screened out for cloud by
applying typical threshold of radar system sensitivity profile, S1 and S5. In addition to this, in case of Figure 1c,
contiguous set of four reflectivity profiles have been considered for computing running mean and standard
deviation. The method followed to generate Figure 1c is the main objective of this paper and is outlined by the
flowchart in Figure 6. This method will be explained below and results and discussion section contains its thorough
information. In this case, insect reflectivity values are similar to those of the cloud but their altitude levels are
significantly different. The contribution due to biota can therefore be removed by S5 curve thresholding and leaving
the contribution due to clouds untouched (Figure 1d). Thus, for the simultaneous presence of cloud and biota echoes
at around same altitude this NER method fails to identify the contributions separately. This NER method also fails
whenever there exist sharp reflectivity changes, usually seen with cloud boundaries/edges. This issue therefore
demands the development of a robust algorithm that explores the fundamental difference between cloud and biota
returns so that it could be identified and separated out these factors automatically.
In order to make the algorithm more robust for running it automatically, a close re-inspection of Figure 1b
infers that cloud returns are much more regular and near homogeneous when compared to biota's returns, which
appears to be spurious or intermittent in occurrence. Therefore, the NER criterion works reasonably well for the case
of homogeneous, isolated stable cloud layers but its robustness will be in question whenever there are vigorous and
quick changes associated with cloud edge and/or structure (will be explained in the discussion of cloud 1-2 in Figure
5). An additional criterion makes the current algorithm robust for complete revival of cloud information from the Z
observations by utilizing the de-correlation periods of biota (close to 3-5 sec). During this time interval significant
changes are not seen within the cloud. To explore this fact, in the next section the same weak low level cloud case
has been chosen further to understand the coherence period associated with cloud and biota.

**3.0      Results and Discussions**

Figure 2 takes the same case as in Figure 1 but confined below 4 km and 80-300 s, (left panel).The added
new NER curves in gray color (S04,S14 andS54; The range correction for the point clear-air target (confined below
3 km) with the start range sensitivity value of reflectivity, i.e., $r^4 x Z_{start\ range}$, where r is range and Z is reflectivity, for
S14, Z is -60 dBZ,  for example). Figure 2 reveals three main type of radar signal region namely (1) consistent radar
returns characterized by the smooth and gradual change(s) associated with cloud particles (at ~ 3.7 km height), (2)
sharp (gradient) and spurious radar returns (at altitude below 2.7 km) due to point target(s) and  (3) receiver noise

floor. In order to locate the above signal types easily, various sensitivity or NER (i.e., S0-S5) curves have been utilized. The second type of signal is associated with a characteristic point target (which has sharp reflectivity gradient feature due to the target's limited spatial as well as temporal spread associated with the radar scattering volume). The third type, noise floor (not radar echo but signal generated in the receiver chain of the radar), is seen to be confined mostly in between S0 and S2. The right panel in Figure 2 corresponds to HTI plot where the echo texture pertinent to the above mentioned three echo types can be clearly visualized. The cloud echoes spreads in the altitude region of approximately 300 m (3.6-3.9 km) with consistent smooth and gradual evolution with its weakest and/or broken structure during 165-190s. In contrast to this the observed irregular point or rounded texture of biota echo spread is seen to be limited temporally around 3-7 seconds and spatially within two (four) range gates (range samples) size (i.e., < 100 m) with strongest reflectivity at its center. This indicates that one second temporal resolution might be good enough to see the biota as point or rounded echo texture. When biota density is more in the lower altitude levels, it is difficult to clearly identify the boundary of one point target from another. Such a scenario, though rare, can lead to misidentification as clouds. The coexistence of cloud and transient high density flocks of biota adds complexity which becomes almost impossible to discriminate. However, this issue is observed to be rare and limited to lowest altitudes only.

To investigate the similarities and contrasting features associated with various contributions to the cloud reflectivity profile, it is important to explore further the case of Figure 1. Statistical echo coherence periods associated with three types (cloud, biota and noise) have been computed for their identification and separation. Both the cloud at ~3.7 km narrow region and biota returns below ~ 1.5 km in Figure 3 are evident above the maximum noise level. Both cloud and biota parts of the Z profiles are expanded to allow for review of the mean (Figure 3b and 3d) and standard deviation (S.D or $\sigma$; Figure 3c and 3e) of Z for every set of consecutive 15 profiles. Figure 3b shows the patterns of the seven mean cloud reflectivity profiles are organized and more consistent or correlated to one another during 105 seconds, this is in comparison to less organized reflectivity profiles due to biota that are much less consistent or correlated with one another in figure 3d. Moreover, the corresponding seven $\sigma$ profiles show differences for cloud that is less than 1.5 $\sigma$ (figure 3c). By comparison differences in profiles due to biota are more than 4.0 $\sigma$ most of the time (figure 3e). It is seen that the mean cloud reflectivity peak values gradually extend from 3.7 to 3.8 km where the corresponding standard deviation values are less than 1$\sigma$. In order to further test the minimum de-correlation time associated with cloud and biota, the averaging time is reduced to a set of 5 profiles (5 sec) with the same data (see Figure 4). In this case also, Figure 4c depicts $\sigma$ for all the seven mean cloud reflectivity profiles are below 1.5 dBZ with peak <1$\sigma$. This manifests that volumetric distribution nature of cloud particles is statistically more homogeneous or show less dispersion. However, Z values associated with biota show random behavior with significant dispersion >1.5$\sigma$ dBZ (Figure 4e). This high dispersion in the Z values infers that the echo due to biota de-correlates quickly within ~5 second time interval (see Figure 4d-4e). It is seen from Figure 3 that for vertical levels from 0.9 km to 1.5, the sharp peaks in reflectivity profiles and strong dispersion of > 3$\sigma$ dBZ are associated with the return from biota. This is attributed mostly to the observed intermittent point target nature of biota echoes plausibly due to the rambling or meandering motion of biota within the radar sampling volume.

Moreover, the inherent radar system noise (random in nature) dispersion is observed to be in between the cloud and
biota (1.5-3.0 σ dBZ). It is evident from the top panels of Figure 3-4 that cloud reflectivity profiles show relatively
consistent trend and correlation among the contiguous mean profiles computed from the set of 15  Z profiles than
computed from the 5 profiles. This may be mainly due to the homogeneities or in-homogeneities associated within
the chosen data sets those are independent to one and another. Therefore, in order to preserve the real time sequence
of observations for the study of cloud evolution as well as to recover underlying smooth trends pertinent to natural
clouds, a four-point moving or running average is applied on the time series of Z data instead of deriving a simple
average. The four seconds is the optimal moving average time for yielding the best cloud results (Figure 5) by
characterizing the cloud to biota echoes coherent to incoherent property during the moving average period.  By this
four point running average, biota echo become incoherent due to its short de-correlation period (~4 sec) whereas
those echoes de-correlating over longer periods indicate the presence of clouds. To understand the degree of
dispersion, along with σ the absolute deviations in mean and median values have also been analyzed.  Their relation
with σ is seen to be as mean absolute deviation slightly smaller than σ as σ/1.253 where as median absolute
deviation smallest as σ/1.483. This work makes use of the statistical mean and σ but using above relation one can
relates the present results with other statistical central tendencies of data distribution.  Next, the filtering of noise and
biota from the presence of cloud using the cloud radar reflectivity profile will be explored. The segregation has been
carried out using theoretical radar echo sensitivity curves and statistically computed echo de-correlation periods and
finally tracking the cloud echo peak to its adjacent sides till it is close to the S1 profile for the cloud height. The
above set of tasks, Theoretical Echo Sensitivity and observed Echo based Statistics for cloud height Tracking
(TEST), is repetitively performed on the cloud radar Z measurements under an algorithm whose flowchart can be
seen in Figure 6. The algorithm used in this work is named as TEST and can be summarized below:
1.  Wherever the moving mean Z values in the profile are equal to or above the S5 can be qualified as cloud or
biota echo. This step ensures removal of the system noise floor.
2.  Those altitude regions of the qualified echo are then further scrutinized to identify clouds using the
minimum thickness of greater than 100 m (to strictly avoid biota that are found to extend less than 2 range
gate each of  50 m) and mean standard deviation below 1.5σ dBZ.
3.  In order to keep the identified cloud's structure, intact, the identified cloud peak(s) are tracked back on
either side (towards upper and bottom heights) up to around (preferably 1-2 dBZ) the mean noise profile
S1.
4.  In order to remove the isolated echo floor, those are probable not cloud but the existence is due to the
abrupt disconsolation at the subsequent running average by the restrictions of step 2, frequency count of Z
profile has been constrained as height levels where the Z frequency count falls below 5% of total
measurement duration used to drop those isolated echoes.

First two steps ensure the identification and removal of non-hydrometeor contributions from the cloud radar
reflectivity profile which can then be used for inferring unbiased vertical cloud structure. However these two steps
are insufficient for recovering the weakly echoing cloud boundaries associated with the sharp reduction in cloud
droplet size and concentrations. For having intact cloud height information(step 3), identified cloud echo peak(s)
needs to be backtracked along the either sides on the reflectivity profile till its value falls close to the mean noise
floor for radar receiver.  It is interesting to note that the cloud echo regions are always stronger and above the mean
noise fluctuations i.e., S1.  Therefore at the left side of the curve, S0 to S1, always appears as a void region in the 2-
dimentional reflectivity plot wherever there is a presence of cloud, no matter weak or strong (just below 4 km in the
left panel of Figure 1 and 3). This causes sharp boundary gradients between cloud and noise in the vertical profiles
of Z and hence with the corresponding $\sigma$. This can be used as a visual criterion for detection of cloud.

Figure 7 is similar to Figure 1 but it represents a multi layer pre-monsoon cloud system for the period 1200-

1205 UT, 29 May 2014. Various labeled altitude regions (biota, noise and cloud) of the vertical reflectivity structure
show typical mean features that can be broadly classified the returns into cloud and non-cloud (biota and noise)
portion. Furthermore, Figure 7 shows the typical variety of cloud layers existing within the vertical structure of
tropical cloud as well as morphological features pertinent to pre-monsoon thunderstorm activity. The cirrus layer at
12-14 km shows gradual structural change having peak reflectivity values of ~ 5 dBZ. Here, the high reflectivity
values contribute to form single deep convective cloud by merging with the cloud layer that exists at lower heights.

Figure 8a and 8b reveal the reflectivity time series associated with the labeled non-cloud and cloud portion

of Table 2 respectively. Noise and biota shows max 2 dBZ fluctuations around the 4-point-running mean reflectivity
whereas for biota the max fluctuation is 3-5 dBZ (bold solid line).  It can be understood that noise values increase
gradually with altitude with $\sigma$ values ~ 2.3 whereas sharp boundary gradients associated with biota and ragged
shallow cloud regions (cloud 1&2 in Figure 7) also show higher $\sigma$ values > 3 dBZ. Stable or layer cloud regions
(cloud 4 & 5 in Figure 7) show significantly standard deviation below $2\sigma$ (dBZ). Further, it is interesting to examine
the time series plots for the contrasting variations between the biota and noise and cloud regions with Figures 8a and
8b. The range of dBZ variability is 4-10 for biota and 2-4 for noise and for cloud that is less than 1 within an interval
of 5-10 seconds. The corresponding variability in standard deviation (S.D) is observed to be 4-10 $\sigma$ for biota, 1.5-3.5
$\sigma$ for noise and ~ 1 $\sigma$ for cloud (<1 $\sigma$ for cloud peak) except for weaker cloud regions.  These statistical
characteristics of all types of observed cloud echoes have been tabulated in the Table 2.

Figure 9 demonstrates the application of the work presented here and illustrates the significant differences

between the uncorrected (Figure 9a) and corrected (Figure 9b) reflectivity profiles.  The peaks in frequency
distribution of uncorrected cloud reflectivity profiles at just below -50 dBZ, in between -50 and -40 and just above -
40 dB are the predominant contributions from noise (middle panel of Figure 9a).  These noise regions bias severely
the corresponding histogram frequency distribution at three different altitude levels that are associated with the
Johnson's tri-modal cloud distribution (extreme right panel of Figure 9a).  In order to infer the distribution of cloud
reflectivity values in the various altitude regions pertinent to tri-modal cloud vertical structure  (Johnson et al.,
1999), the observed vertical structure is subdivided into warm or low (<3.6 km), mixed or mid (3.6 km $\geq$ altitude
$\leq$8.6 km) and ice or high (>8.6 km) phase and/or level clouds. The plots of uncorrected reflectivity distribution
clearly shows skewness towards lowest values of reflectivity (below -50dB, -40 dB and -30 dB for low, mid and
high level respectively seen with right panels of Figure 9a). This is mainly due to the predominance of noise
contribution except for the low cloud regions where the contribution of biota is also included. After applying the
TEST algorithm the corrected reflectivity distribution peaks at -42dB, -35 dB and -22 dB for low, mid and high level
respectively (right panel of Figure 9b) reflects the actual scenario of the cloud system. This method is simple and
has potential to bring out the statistically significant micro- and macro-physical characteristics from meteorological
information (i.e., cloud) and hence for better characterization of the cloud vertical structure over a region.
In order to test the merit of the current algorithm on filtering out the non-hydrometeor contribution with Z
profile, the parametric thresholds on Pulse-Pair (PP) processed Z and few polarimetric variables profiles of the cloud
radar measurements have also been considered in place of usual Fast Fourier Transformation (FFT) process. The
FFT process is capable to provide only polarimetric parameter, i.e., linear depolarization ratio (LDR). Figure 10 is
similar to the Figure 1 that illustrates FFT (top) and PP (bottom) processed Z profiles on 28 Aug 2014 but are 15
minutes apart from one another (0415 and 0400 UT respectively) which causes some dissimilarities in the observed
three layer cloud structure between the two plots (upper and lower panel). Minimum range of the noise floor in the Z
profiles (2-D plot in the first panel) is seen to be grater for PP than FFT processing. The TEST algorithm performs
in a similar way for both the FFT and PP processed Z profiles and is able to isolate the cloud structure as best as
possible.   Figure 11 explores further the polarimetric capability of the KaSPR in separating out the
meteorological/hydrometeor contribution with Z by using critical threshold on the PP-polarimetric measurements
that correspond to the bottom panels of Figure 10. The top panels of Figure 11 stand for HTI plots of, three
polarimetric parameters namely, LDR, $\Phi_{dp}$ and $K_{DP}$. Computation of LDR is inherently limited to the cross polar
isolation of the radar system that is -27 dB for KaSPR. Hence, high LDR values above -17 dB are mostly seen with
biota and low LDR values below -17 dB are seen with cloud. Low to lower LDR values (i.e., <-17 dB to -25 dB) are
strictly confined within the peak values of co-polar reflectivity (> -10 dB) of cloud altitude regions, ~ 8-10 km.
Except the inherent limitations associated with LDR, these results are in agreement with earlier reported results (e.g.
Bauer-Pfundstein and Görsdorf, 2007 and Khandwalla et al., 2003). The LDR, $\Phi_{dp}$ and $K_{DP}$ threshold values are set
below -17 dB, $56^0$ and $-15^0$ km$^{-1}$ respectively, can be used to filter out biota from the corresponding Z profiles that
are shown at lower panels of Figure 11. The threshold used for $\Phi_{dp}$ and $K_{DP}$ are subjective depending on the
observed case for better filtering of biota. These polarimetric threshold methods are although successful in filtering
out the non-hydrometeor contributions but they are bound to sacrifice the weaker portion of the cloud where
polarimetric computations are not perfect. Thus, polarimetric method is incapable to preserve the weaker portions of
the whole cloud regions where the TEST method is noticed to perform better (bottom right panel of Figure 10). This
further proves the efficiency of the proposed TEST method. This has implemented in the post-processing of high
resolution reflectivity measurements. The method developed here is far simpler and provides a superior solution to
filtering out signal due to noise and biota and preserve cloud data in the form of pure hydrometeor reflectivity
measurements which can be used to infer the true characteristics of clouds.

Figure 12a demonstrates further application of the current work on filtered cloud reflectivity profiles
(bottom plot) by considering the six hours evolution of variety of tropical cloud systems. On 21 May 2013, a typical
convective cloud system present during pre-monsoon season was observed. This event is composed of three
systems, first three hours (00:00-03:12 UT) shows stratiform cloud confirmed from bright band occurrence at an
altitude of 4 km AGL, convective system around 0500 UT, which is a cumulus congestus initially , and above it
cirrus (ice) cloud in the altitude range of 13-14 km. The screened out reflectivity profile can therefore be utilized to
fully characterize the tri-modal cloud episode as shown in Figure 12b. The mean reflectivity profile with standard
deviation bars reveals the nature of important phase change regions associated with cloud vertical structure. The
change in cloud processes in the cloud vertical structure is closely associated with the phase of cloud water that is
strongly linked with the predominant change of temperature.

Finally, Figure 13 and Figure 14 are cases of much worthy to discuss the merits and demerits of the TEST
algorithm for shallow cumulus clouds present with biota. In fact this is the concluding figure of the work where
besides to the Reflectivity based TEST (first column), LDR (second column) and SW (last column) measurement of
the same cloud radar are also considered. Second row panels in figure 13 are differing from first only by filtered out
for noise using sensitivity curve S5 and to allow cloud and biota presence with the radar measurements. The higher
level biota is noted to be much organized just above 2.5 km. Shallow ABL cloud regions show LDR values <-20 dB
whereas insects shows varied LDR values in the range of -25-to -5 dB. Thus, LDR alone is not sufficient to remove
all insects (figure 13e). Smaller echo coherence period associated with biota are further confirmed with less spectral
width values (<0.3 $m^2$ $s^{-2}$; figure 13f). Higher spectral width values, of the order of ~ 1 $m^2$ $s^{-2}$ of the cloud indicates
the random motion of the smaller particles of cloud within the radar scattering volume are affected by the ABL
turbulence. The discussed TEST algorithm (fig 13g) is able to screening out the cloud and filter out the biota part
significantly. Further, TEST fails to isolate relatively stronger biota returns exits within the cloud due to the missing
of strong reflectivity gradient (both in short intervals of height and time scale) which fails to give needed high
standard deviation values to filter out those. In order to ensure those as biota and then to isolate those returns, the
LDR values larger than -14 dB and SW values much smaller than 0.5 $m^2s^{-2}$ have been chosen here. Identified
isolated biota returns outside the cloud by TEST and the above critical thresholds with LDR and SW are found to be
similar significantly excepted at few places. It infers that, using threshold value alone either with LDR or SW
measurements threshold value fails to filter out all biota returns due to either persistent low LDR or high SW values
associated with those biota. However, it can be seen with figure 14 (similar to figure 13 but a typical case of high
number density of biota noticed on 10 Sep 2013 during 0738-0742 UT) that TEST alone unable to remove biota
(figure 14g) but using LDR it becomes much promising (figure 14f). Furthermore, in case of weakly turbulent cloud
portions, they posses near comparable lower SW values as that of biota, under such condition it is complicated to
screen out clouds using SW along (see figure 14i). Similar way, LDR alone is observed to be difficult in filtering all
biota and screen out weak clouds. However, these two diverse and independent radar parameters, Doppler spectral
width and power based polarimetric LDR measurements of KaSPR will be an additional measures on the
identification of cloud to non-hydrometeor echoes of the radar.

It infers from all the above discussions, that the biota presence has been confirmed more than one way by
considering LDR that infers the liquid body presence in the atmosphere (cloud particle, bird or insect), small spectral
width values infers less velocity variance or spread within radar sampling volume. Small velocity variance
associated with biota is obviously due to the sole presence of air-borne biota that usually takes advantage of
dynamics of the atmosphere (initially for flight up by the convective updrafts and later by advection for horizontal
flight at higher levels). Moreover, the velocity spread due to biota is very limited to smaller value than volumetric
small cloud particles those are in general relatively light weight, high in number density and more vulnerable to
small scale local turbulence or entrainment process which gives rise to higher spread or dispersion of velocities to
have high spectral width values observed with cloud particles associated with shallow cumulus cloud. Considering
all these facts, It is interesting to note that the combined TEST, LDR and SW yields best cloud alone results than
any other combination where both cloud and biota co-exists within radar sampling height. Clouds show high spectral
width values ~ 1 $m^2s^{-2}$. Lower spectral width values pertinent to biota infer that velocity variance of scatters within
radar scattering volume is predominantly due to the presence of airborne biota (without much flight maneuver).
This could be the reason to have much smaller time coherence or degree of correlation of Z value with biota is much
smaller (e.g., 4-5 seconds) than clouds. Thus biota echo de-correlation times are small or quicker at the transmitted
pulse scale.  In order to confirm the precise de-correlation periods associated with the observed biota and cumulus
clouds (figure 13a) that are assumed to be vertical radar transact across ABL, simple auto correlation function
(ACF) has been used with the time series data of Z corresponding the biota at 1.59 and 2.66 km and cloud levels at
lower/base, mid and top (single range gate (solid line) as well as averaged to its top and bottom range gate (dashed
line). The ACF's lag, 0-300, correlations for the cloud and biota are clearly seen with figure 15. Thus, from the ACF
analysis it is clear that biota shows quicker (~4 seconds) de-correlations periods than cloud (~ 40-170 seconds).
Moreover, it is interesting to note that single height level (solid line) observations are showing relatively weaker
correlation than averaged (dashed line) one, this is much significantly seen with cloud echoes that confirms that
clouds are have high degree of phase coherence, mainly because of clouds are wide spread (both time and space) in
nature, that becomes additive to have high correlation than single level whereas for quickly de-correlating biota or
random noise there is no much difference between them.  Thus, clouds show varied de-correlation periods above 30
seconds but biota mostly de-correlate very much less than 10 seconds. Hence, the hypothsis proposed for TEST is
proved here with.
**4.0    Summary and Conclusions**
Millimeter-band radars are very sensitive to detect small targets such as cloud droplets and also insects and
other biological particulates (biota) present in great number in the lower atmosphere. Polarization measurement is an
efficient mean to discriminate cloud echoes from non-hydrometeor scatterers that share in common very low
reflectivity. Unfortunately not all radars are equipped with polarization measurements. This paper proposes for these
standard radars a simple technique able to separate meteorological and non-meteorological echoes. It uses only
successive vertical reflectivity profiles acquired by a 35-GHz radar operated at vertical incidence with a 50 m pulse
length and one second temporal sampling. Because of the high spatial and temporal resolution, most of the time only
one or no biota target is present in the pulse resolution volume. In contrast, cloud echo is due to millions droplets
that occupy the pulse volume. As a consequence signal variability at a given range between two vertical profiles is
much more important for biota scatterers than for cloud echoes. Signal variability is given here by the standard
deviation of the reflectivity over the time of four profiles that corresponds to the typical duration of the biota echoes
crossing the antenna beam. The threshold value that separates distinctly biota from cloud is obtained from statistical
analysis of a large radar observation set. Indeed this value should be adjusted for a radar having different
characteristics. This study responds to a real issue for anybody who wants to extract physical quantities from radar
signal. The methodology used is validated with polarization measurements provided by the same radar.

It has been demonstrated that high resolution vertically oriented zeroth moment (reflectivity) measurements
of cloud radar are solely assured to segregate the hydrometeor and non-hydrometeor contributions with it.
Theoretical noise equivalent reflectivity curves are used to remove the system noise and importantly for recovering
the weak cloud boundaries that are very closely hidden within the mean noise floor (curve S1) of the radar system.
The simple statistical variance of continual radar echoes show the contrasting different characteristic of signals like
high dispersion (more than $2\sigma$) is associated with the highly spurious and intermittent echoes of biota and low
dispersion (less than $1\sigma$) is associated with coherent nature of echoes of cloud hydrometeors and for noise it is in
between 1.5 and 3.0 $\sigma$. Furthermore, these characteristic features are mainly holding a key to demarcate the returns
of cloud hydrometeor to those from biota and noise. Running mean and standard deviation of off-line reflectivity
profiles for ~4-5 seconds that works well to filter out all non-hydrometeor returns. In this way, the time coherence of
radar returns at every range sample was checked for every 4 seconds as off-line window period to infer the de-
correlation period associated with biota that show promise in identifying and filtering out the biota returns.  The
proposed TEST algorithm evaluates the observed cloud radar reflectivity profiles with combined theoretical radar
sensitivity curves and statistical variance of radar echo and then tracks the cloud peak at either side to obtain the
complete cloud height profile. In case of azimuth and elevation radar surveillance scans (PPI and RHI, for example),
there is a regular change in the radar sampling area that disables to have exclusive set of measurements required to
perform the TEST method. But this method is advantageous and easily adaptable for better characterization of any
high-resolution vertical profile measurements. The robustness of TEST is also proved through polarimetric and
spectral width measurements and found that that works much better, particularly within the cloud region, at the
cloud radar frequencies. TEST constrained using LDR found much promising under high density biota condition
whereas superior performance of combined TEST constrained with both LDR and SW has witnessed with highly
turbulent shallow convective clouds. Such scrutinized reflectivity profiles have been further utilized to investigate
the important CVS pertinent to the various phases of the Indian Summer Monsoon with the aim of improved
prediction. Hence, the proposed TEST algorithm is able to extract the possible unbiased meteorological cloud
vertical structure information with the cloud profiling radar. This enables carrying out the pragmatically effective
research investigations on the seasonal and epochal tropical cloud characteristics.
*Acknowledgements:* IITM is an autonomous organization that is fully funded by MOES, Govt. of India. The
authors are thankful to Director, IITM not only for his whole hearted support for establishing the radar programme
but also for monitoring and acting as a source of inspiration to promote this research to the next level. The authors
are highly indebted to Dr Ernest Raj, Dr Devara and all those who were involved and helped in setting up the
IITM's Cloud Radar Facility, KaSPR as well as KaSPR design and development which was done at M/s Prosensing,
USA. The radar data supporting this article can be requested from the corresponding author
(kalapureddy1@gmail.com).

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

**Figure Captions**

**Figure 1.** (a) Vertical looking cloud radar measured sample ten reflectivity height profiles on 27 April 2014 during 2303-2308 UT. S0 to S5 are the theoretical noise equivalent reflectivity curves with their respective threshold values in bracket. HTI plot of (b) the same reflectivity profile for the duration of 306 sec (c) screened out reflectivity profile for the receiver noise floor and the biota (insects) using running average constrained with standard deviation (d) constrained with NER (S5).

**Figure 2.** (left) Same as 1(a) but for 220 profiles. Extra NER curves here in gray color (S04, S14 and S54) are computed on the basis of the point target radar equation (i.e., $r^4 x Z_{\text{start range}}$, where r is range and Z is reflectivity, e.g., S04, Z is -68 dBZ) (right) HTI plot of Z profiles. Smoothly varying homogeneous cloud layer is at altitudes of 3.5-3.8 km and sharp, rounded and spurious kind of echoes below 2.7 km are due to biota.

**Figure 3.** (a) Same as 1(a) but for 105 profiles. (b) mean and (c) standard deviation of 15 profiles of Z pertinent to cloud height region (3.5-3.9 km) and (d) and (e) same as (b) and (c) but pertinent to biota height region (0.9-1.5 km).

**Figure 4.** Same as Figure 3 but for total duration 35 sec; the mean and standard deviation profiles are for every 5 second interval.

**Figure 5.** Same as Figure 3 but for total duration 10 sec; the mean and standard deviation profiles are for 4-point-moving average.

**Figure 6.** TEST algorithm flow chart that identifies and filter-out the biota and noise echoes for screening-out the cloud contributions with the Z measurements.

**Figure 7.** (a-c) Same as 1(a-c) but on 29 May 2014 during 1200-1205 UT for the duration of 306 sec. Statistics corresponds to the labels on the Z profile can be seen in Table 2.

**Figure 8a.** Time series of the mean and standard deviation (S.D) of Z for biota (bottom panels) and four noise floor regions as per Table 2. Bold solid lines are the 5-point-running mean over the actual time series data (lines with symbol).

**Figure 8b.** Same as Figure 8a but for the cloud regions as per Table 2.

**Figure 9a.** (Left panel) Uncorrected mean reflectivity profile on 29 May 2014 during 1200-1205 UT superimposed with curves S1 (dashed red line) and S5 (solid green line). Histogram of Z profile (Middle panel). (left three sub panels) for altitude regions of low (<3.6 km), mid (3.6 km>=ht<8.6 km) and high (>=8.6 km). The right sub panels each peak of histogram are mapped on to the corresponding three peaks with the whole vertical structure of Z. This infers the noise clearly suppresses the meteorological information.

**Figure 9b.** Same as 9a but it is corrected by filtering out noise and biota. The correction applied to Z profile allows to pop-up the true meteorological cloud reflectivity distribution.

**Figure 10.** Same as 7 but for vertical looking KaSPR measurements at 0400 UT on 28 Aug 2014 using (top) FFT processing (bottom) 15 minutes prior one using PP processing. PP case will be used further to evaluate the polarimetric algorithm performance.

**Figure 11.** HTI plots of (top panel) LDR, $\Phi_{dp}$ and $K_{DP}$ parameters pertinent to PP processed data of Figure 10 and (bottom panels) biota filtered reflectivity after applying corresponding polarimetric thresholds of the respective top panels.

**Figure 12a.** (Top) Same as Figure 7b (uncorrected) and (bottom) same as Figure 7c (corrected) but integrated for duration of 0000-0630 UT taken at an interval of ~ 15 minutes on 21 May 2013

**Figure 12b.** Same as Figure 9b but excluding middle panel for the corrected Z data of figure 12a.

**Figure 13.** Cloud radar measurements of reflectivity (Z), LDR, Spectral Width (SW) with noise (a-c) and filtered out for noise using S5 curve (d-f), TEST algorithm screened output Z for clouds (g), g + biota filtering using LDR > -14 dB (h), h + SW filter for biota using SW < 0.5 $m^2 s^{-2}$ (i).

**Table 1:** KaSPR specifications

| Radar specifications | value |
|---|---|
| RF output frequency | 35.29 GHz |
| Peak power | 2.1 kW |
| Duty cycle | 5 % max. |
| Pulse widths (selectable) | 3.3 µs (50-13000 ns) |
| Pulse compression ratio | 1:10 (1-100) |
| Range gate spacing (resolution) | 25 (50) m |
| Transmit polarization | H or V-pol linear; Pulse-to-pulse polarization agility |
| Receiver polarization | Simultaneous Co- and Cross-polarization linear |
| Receiver noise figures | 2.8 dB min |
| Sensitivity at 5.0 km | -45 dBZ |
| Tx & Rx loses | 1.15 & 0.3 dB |
| IF output to digital receiver | 90 MHz |
| Antenna diameter | 1.2 m |
| Antenna Beam width | $0.5^0$ |
| Antenna gain (includes OMT loss) | 49 dB |
| First side lobe level | -19 dBi min. |
| Cross-polarization isolation | -27 dB |

**Table 2:** Statistical mean and standard deviation of cloud radar reflectivity corresponds to the selected height regions, which are labeled, on the Figure 7.

| Label | Mean Z for 305 sec (4 sec) dBZ | σ for 305 sec (4 sec) |
|---|---|---|
| Biota (1.2-1. 7 Km) | -54.1(-55.0) | 4.08 (3.4) |
| Noise 1 (2.1-2.4 Km) | -52.9 (-52.4) | 2.33 (1.9) |
| Noise 2 (5.9-6.2 Km) | -44.4 (-44.2) | 2.22 (2.3) |
| Noise 3 (11.1-11.6 Km) | -39.1 (-39.1) | 2.30 (2.2) |
| Noise 4 (14.7-15.2 Km) | -36.7 (-36.9) | 2.29 (2.2) |
| Cloud 1 (3.7-3.9 Km) | -36.2 (-28.3) | 5.99 (12.7) |
| Cloud 2 (4.8-5.1 Km) | -31.8 (-22.7) | 5.54 (4.5) |
| Cloud 3 (6.8-7.2 Km) | -0.4 (0.3) | 2.60 (3.5) |
| Cloud 4 (9.8-10.2 Km) | -10.9 (-9.9) | 2.03 (3.1) |
| Cloud 5 (12.8-13.2 Km) | 3.1 (1.4) | 0.86 (1.0) |

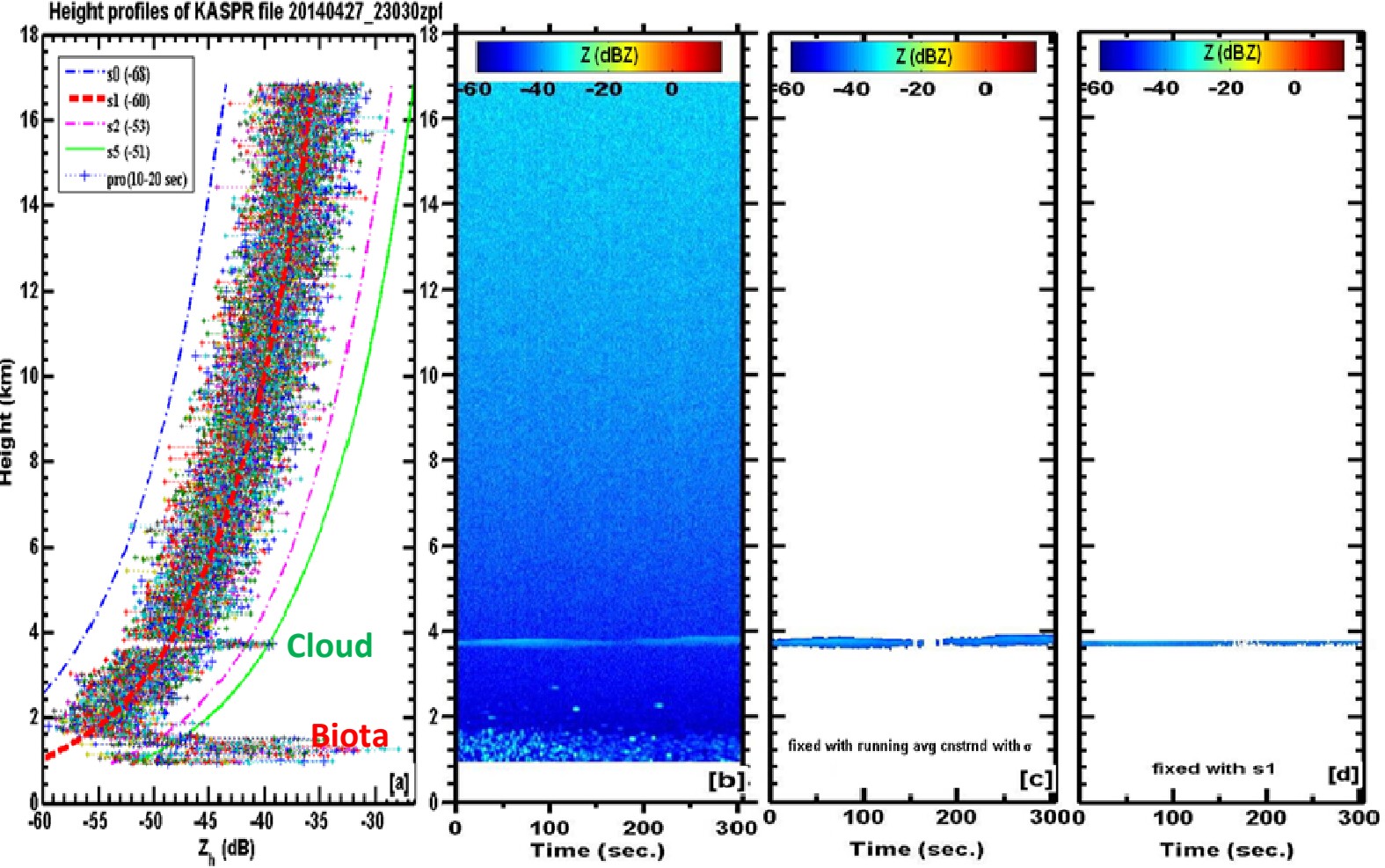

**Figure 1:** (a) Vertical looking cloud radar measured sample ten reflectivity height profiles on 27 April 2014 during 2303-2308 UT. S0 to S5 are the theoretical noise equivalent reflectivity curves with their respective threshold values in bracket. HTI plot of (b) the same reflectivity profile for the duration of 306 sec (c) screened out reflectivity profile for the receiver noise floor and the biota (insects) using running average constrained with standard deviation (d) constrained with NER (S5).

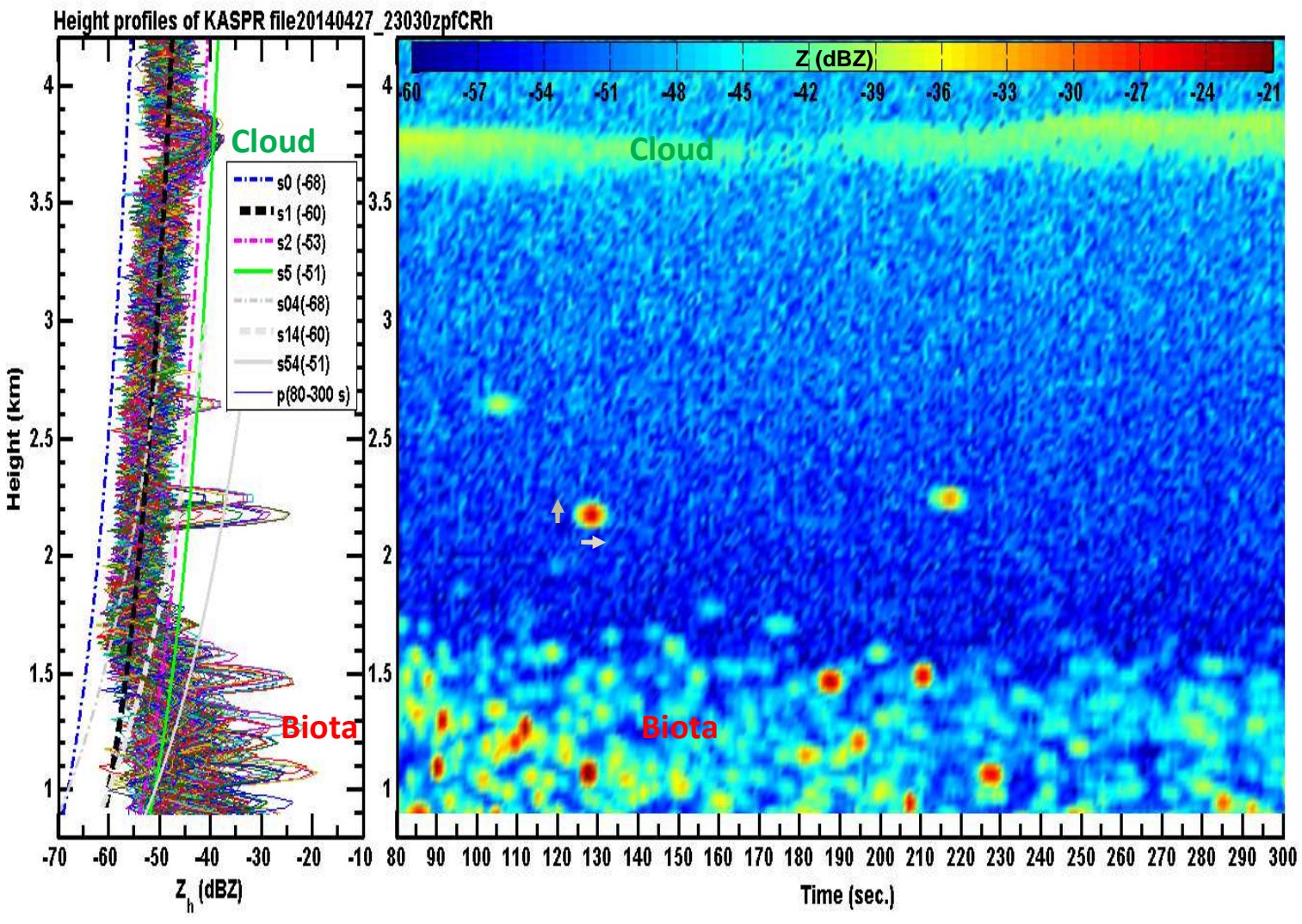

**Figure 2: (left)** Same as 1(a) but for 220 profiles. Extra NER curves here in gray color (S04, S14 and S54) are computed based on the point target radar equation (i.e., $r^4 \times Z_{start\ range}$, where r is range and Z is reflectivity, e.g., S04, Z is -68 dBZ). **(right)** HTI plot of Z profiles. Smoothly varying homogeneous cloud layer is at altitudes of 3.5-3.8 km and sharp, rounded and spurious kind of echoes below 2.7 km are due to biota.

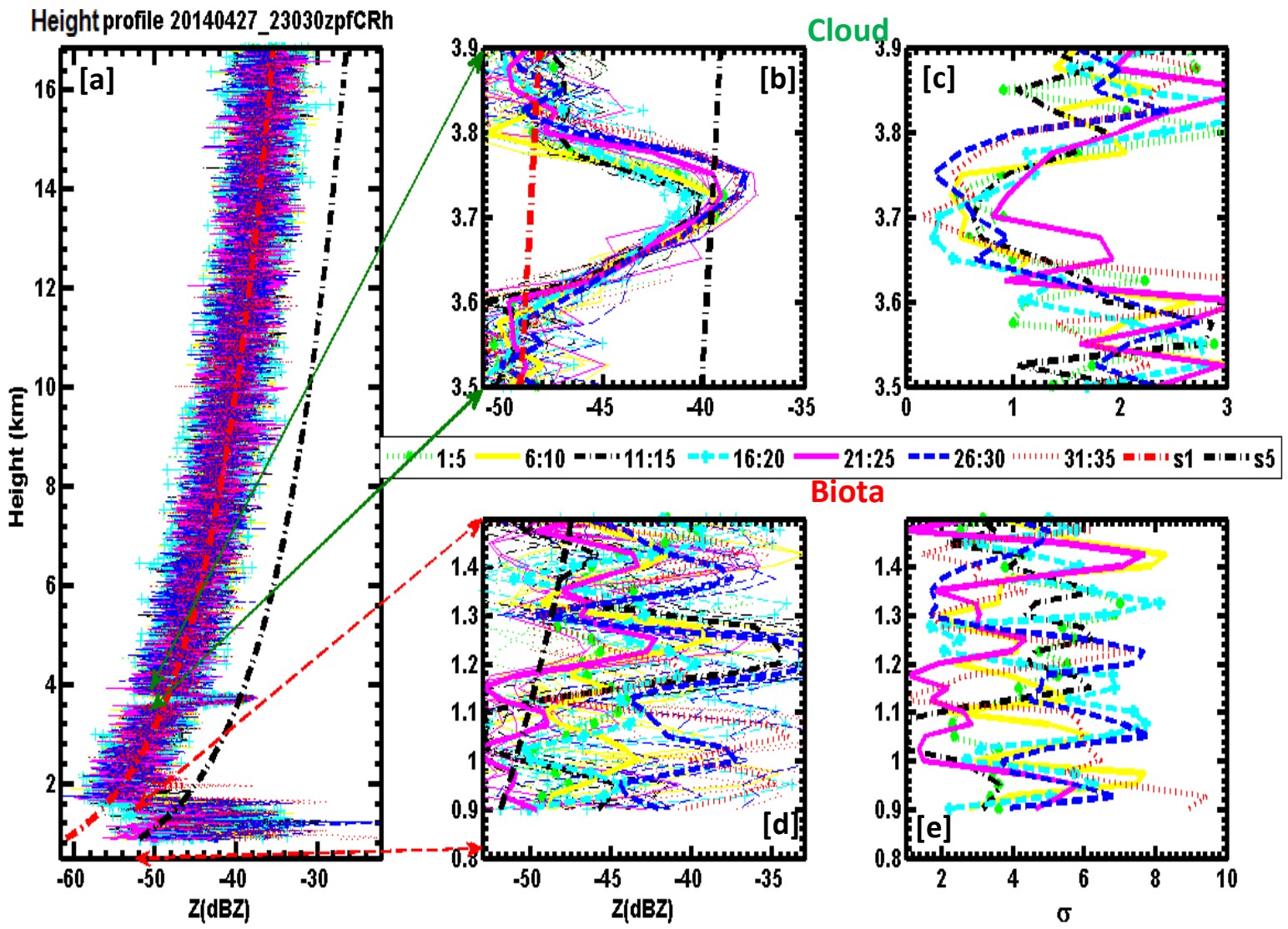

**Figure 3: (a) Same as 1(a) but for 105 profiles. (b) mean and (c) standard deviation of 15 profiles of Z pertinent to cloud height region (3.5-3.9 km) and (d) and (e) same as (b) and (c) but pertinent to biota height region (0.9-1.5 km).**

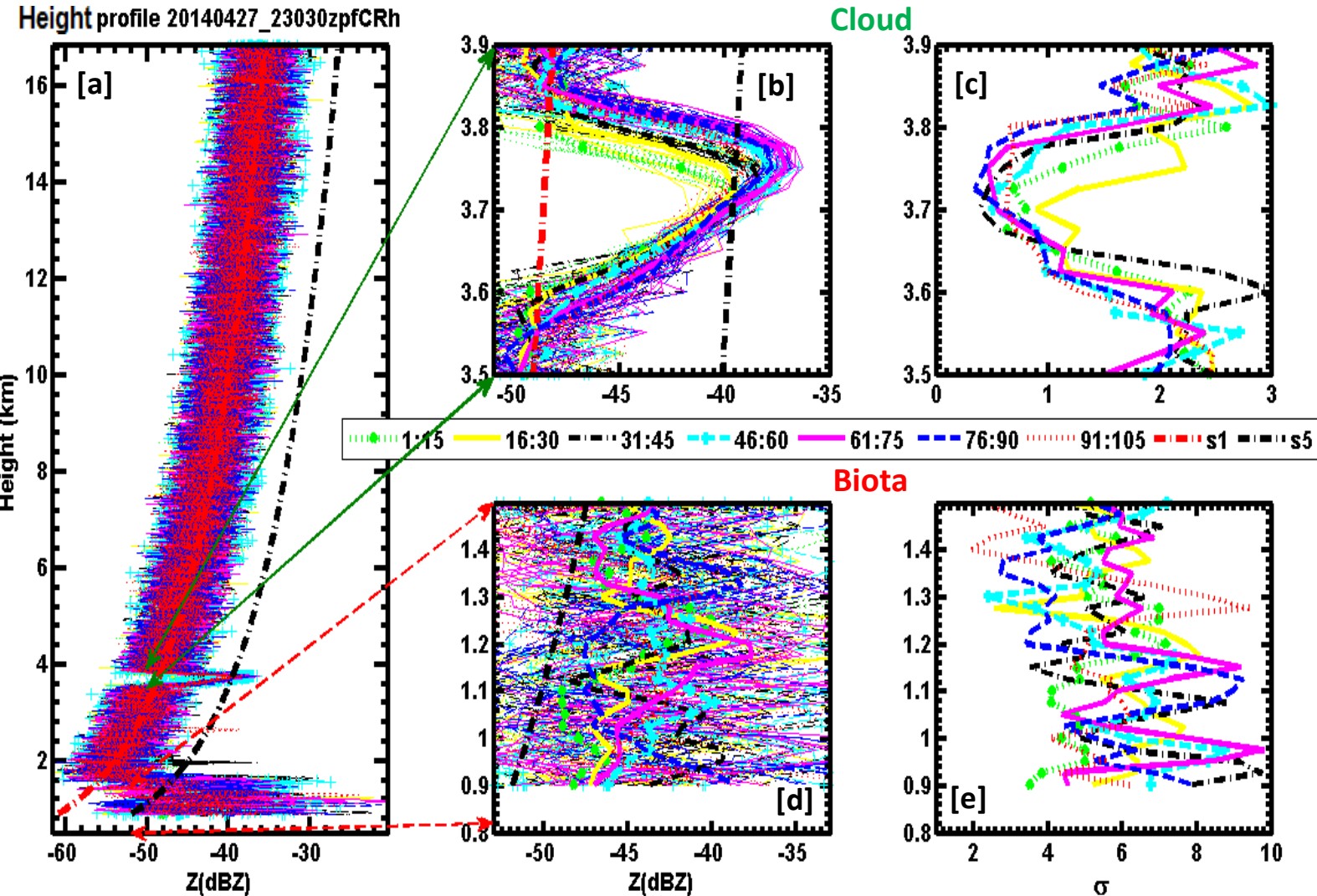

Figure 4: Same as Figure 3 but for total duration 35 sec; the mean and standard deviation profiles are for every 5 second interval.

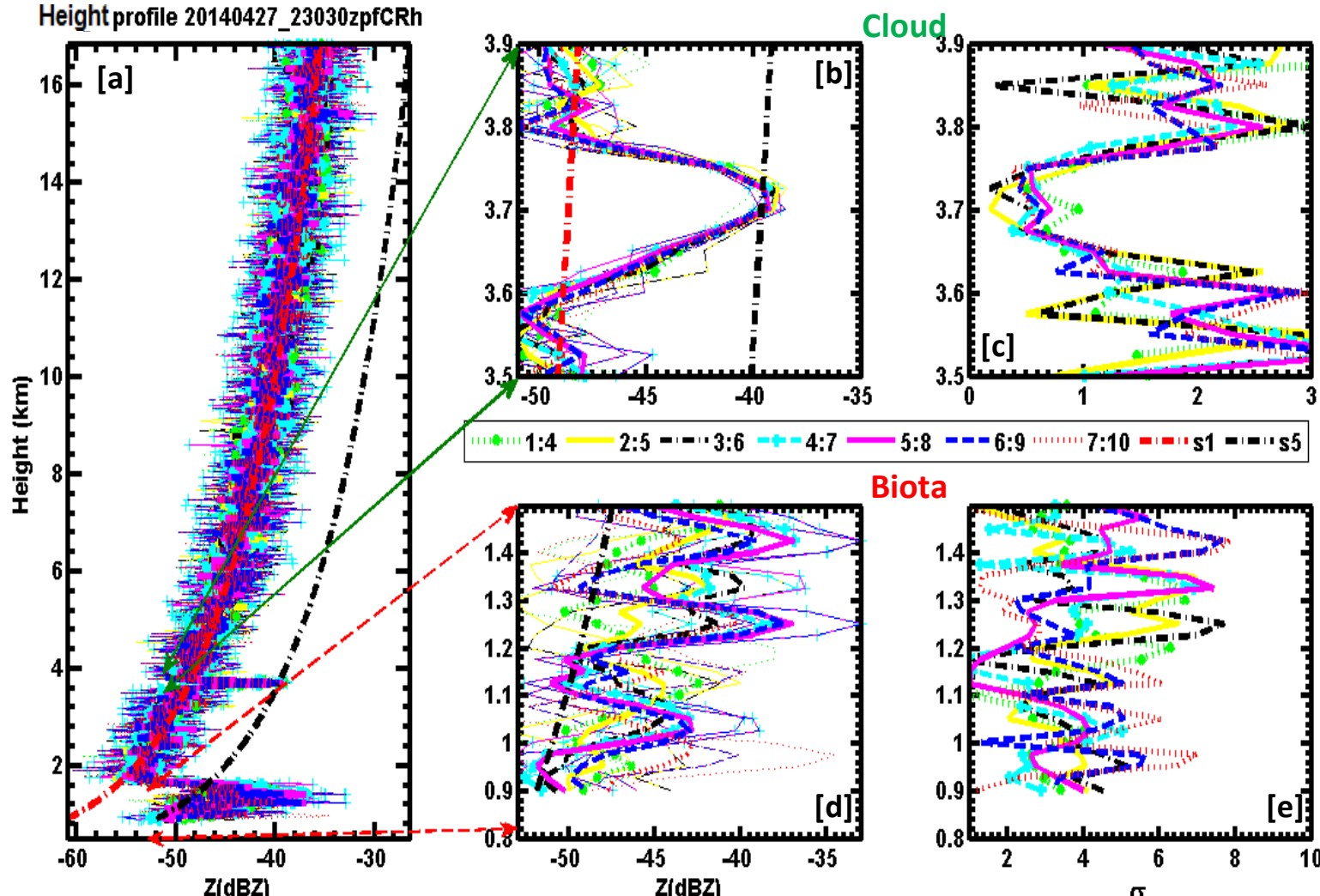

**Figure 5: Same as Figure 3 but for total duration 10 sec; the mean and standard deviation profiles are for 4-point-moving average.**

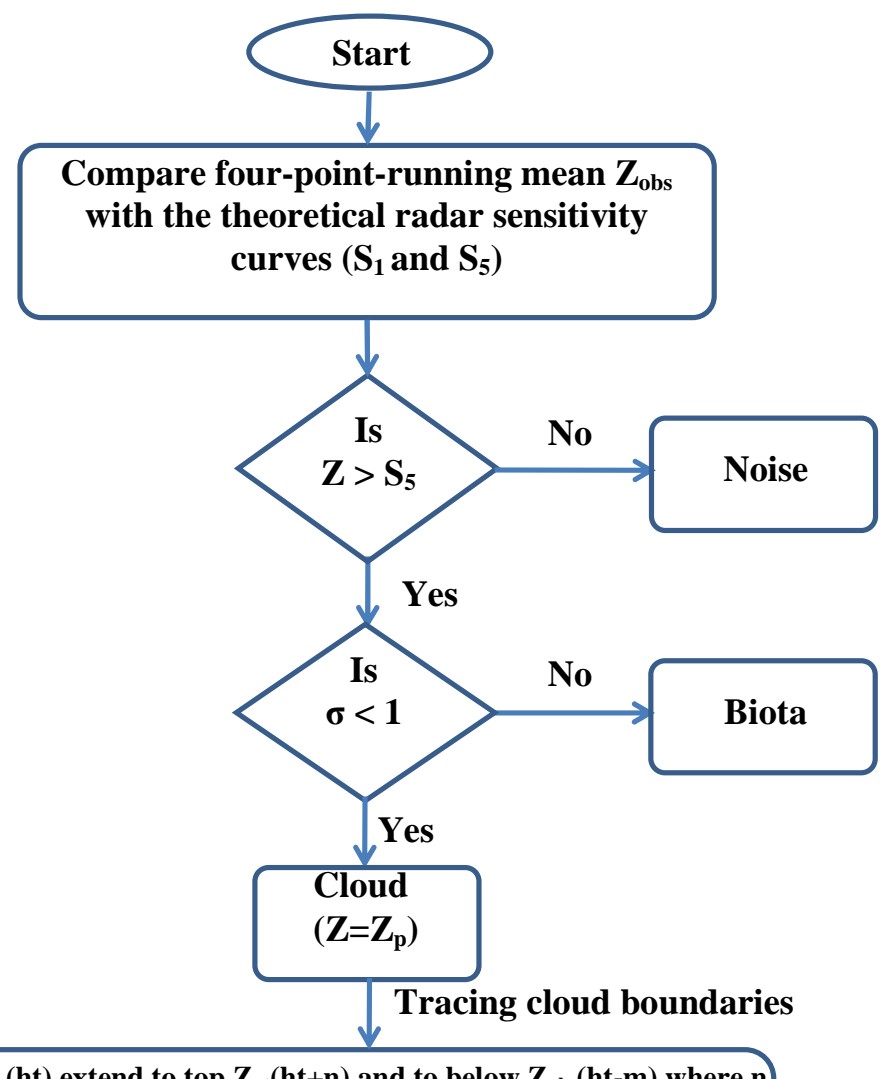

**Figure 6: TEST algorithm flow chart that identifies and filter-out the noise and biota echoes for screening-out the cloud contributions with the Z measurements.**

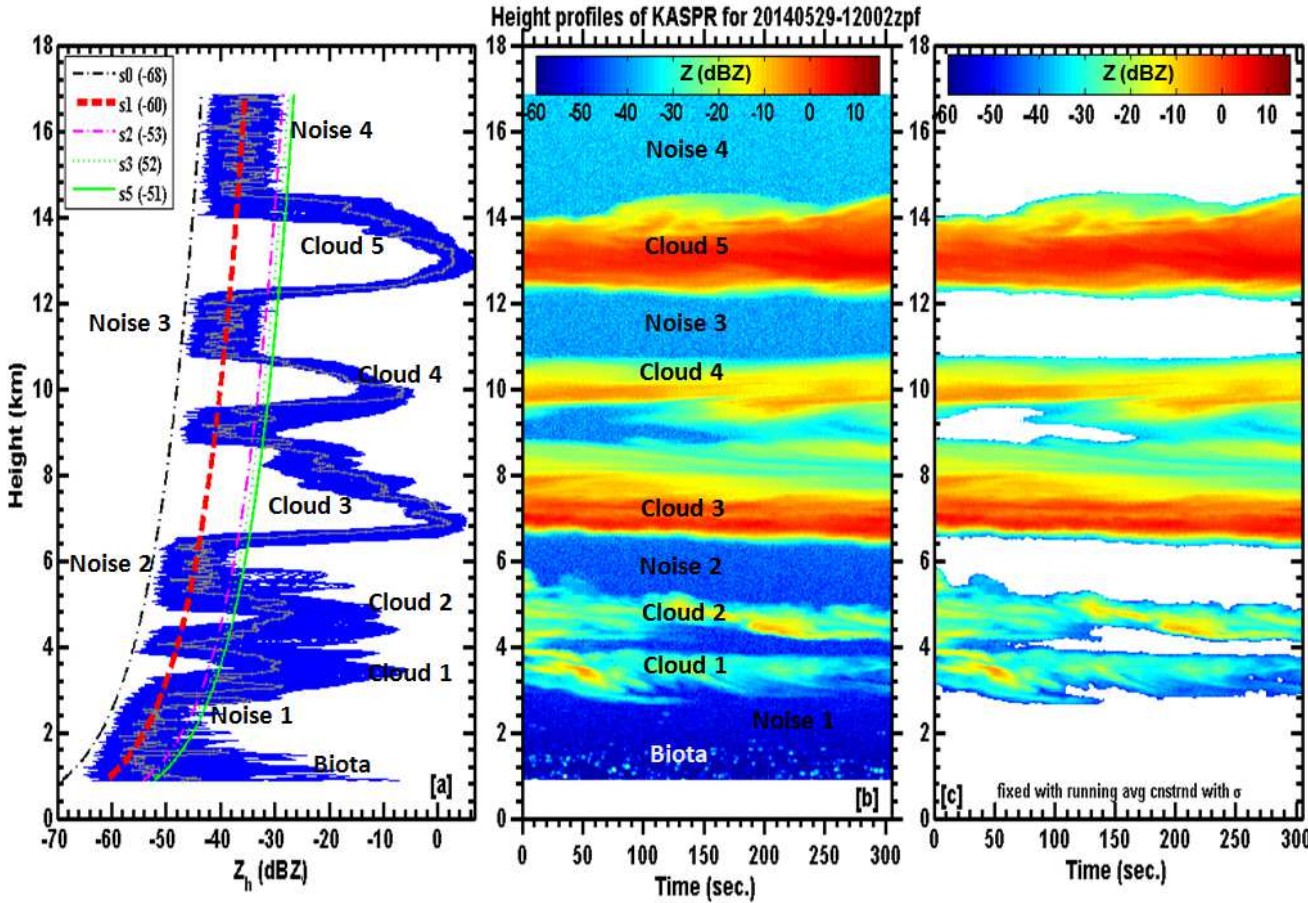

**Figure 7: (a-c)** Same as 1(a-c) but on 29 May 2014 during 1200-1205 UT for the duration of 306 sec. Statistics corresponds to the labels on the Z profile can be seen in Table 2.

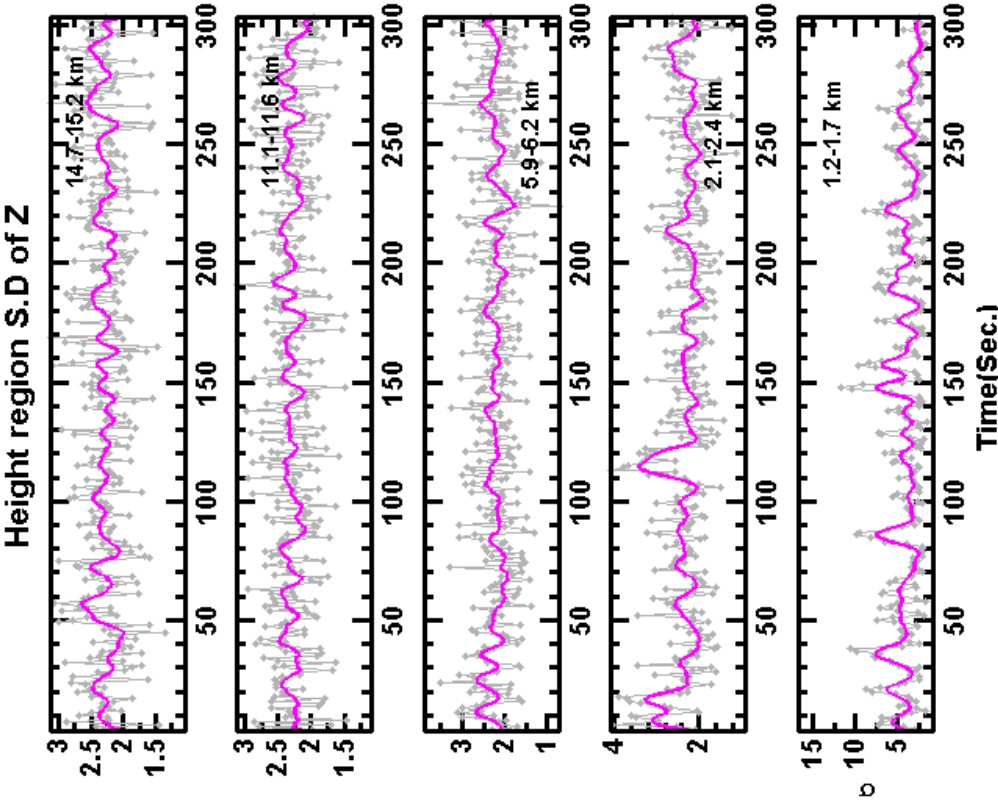

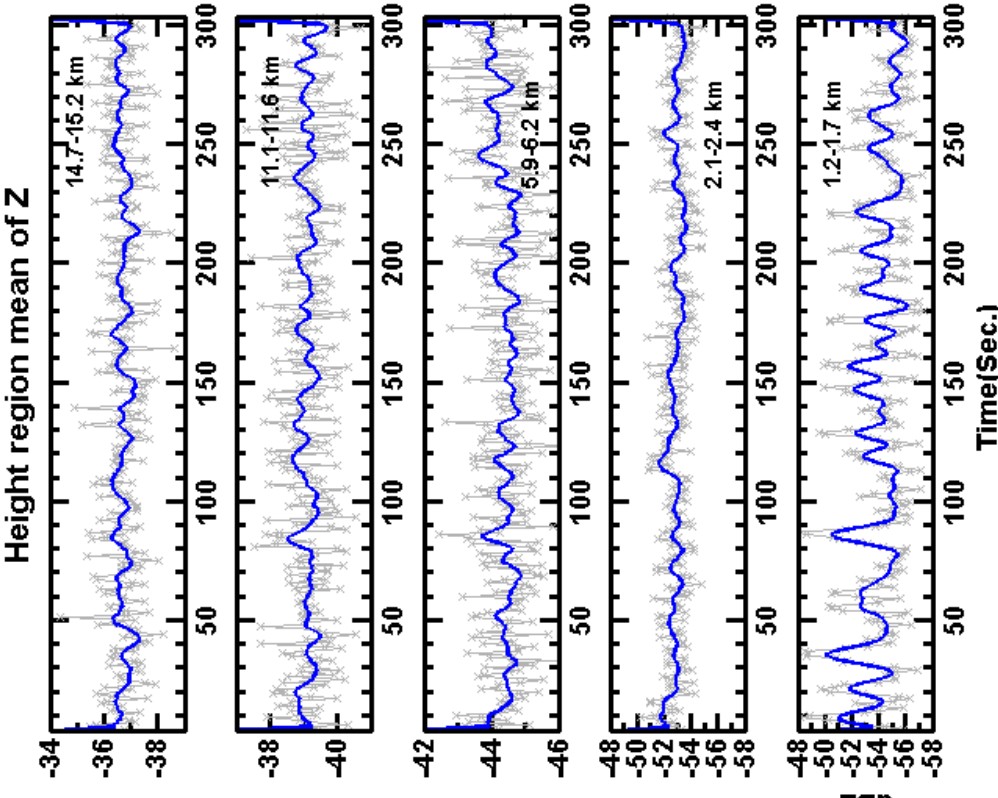

**Figure 8a: Time series of the mean and standard deviation (S.D) of Z for biota (bottom panels) and four noise floor regions as per Table 2. Bold solid lines are the 5-point-running mean over the actual time series data (lines with symbol).**

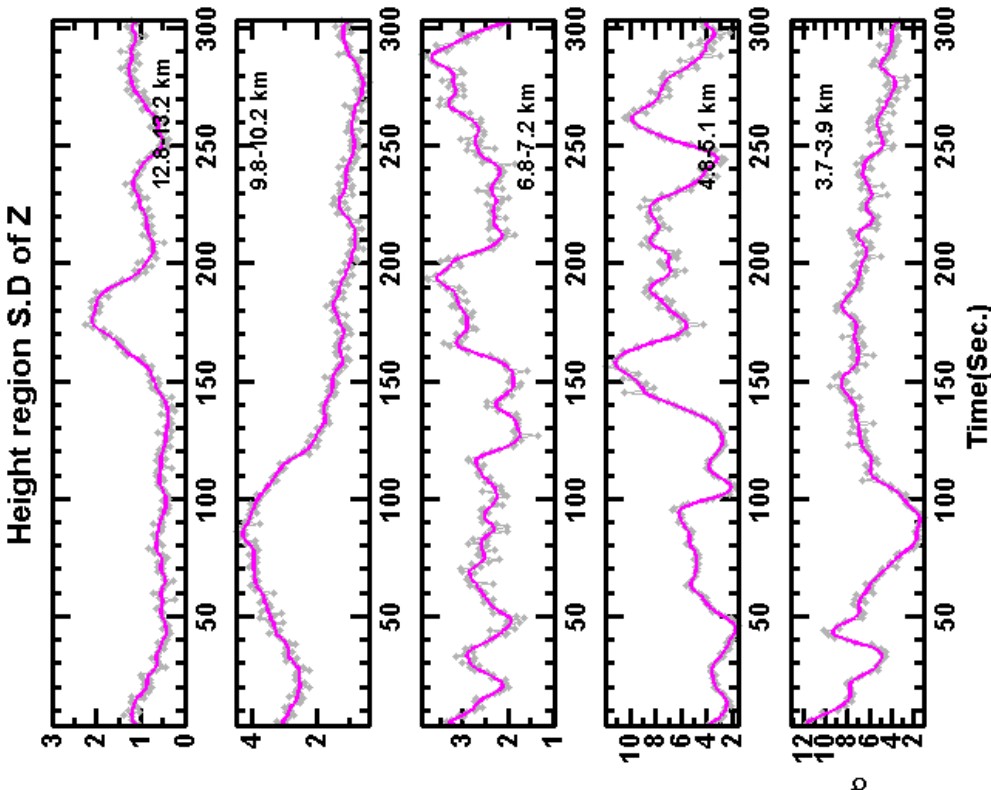

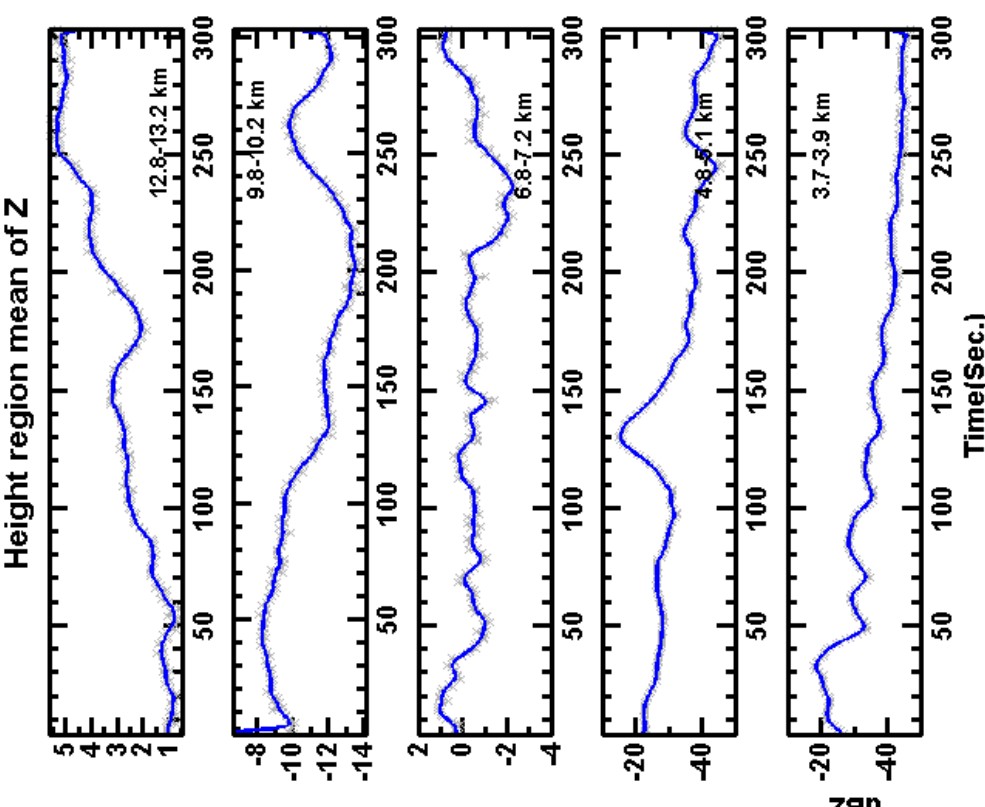

**Figure 8b: Same as Figure 8a but for the cloud regions as per Table 2.**

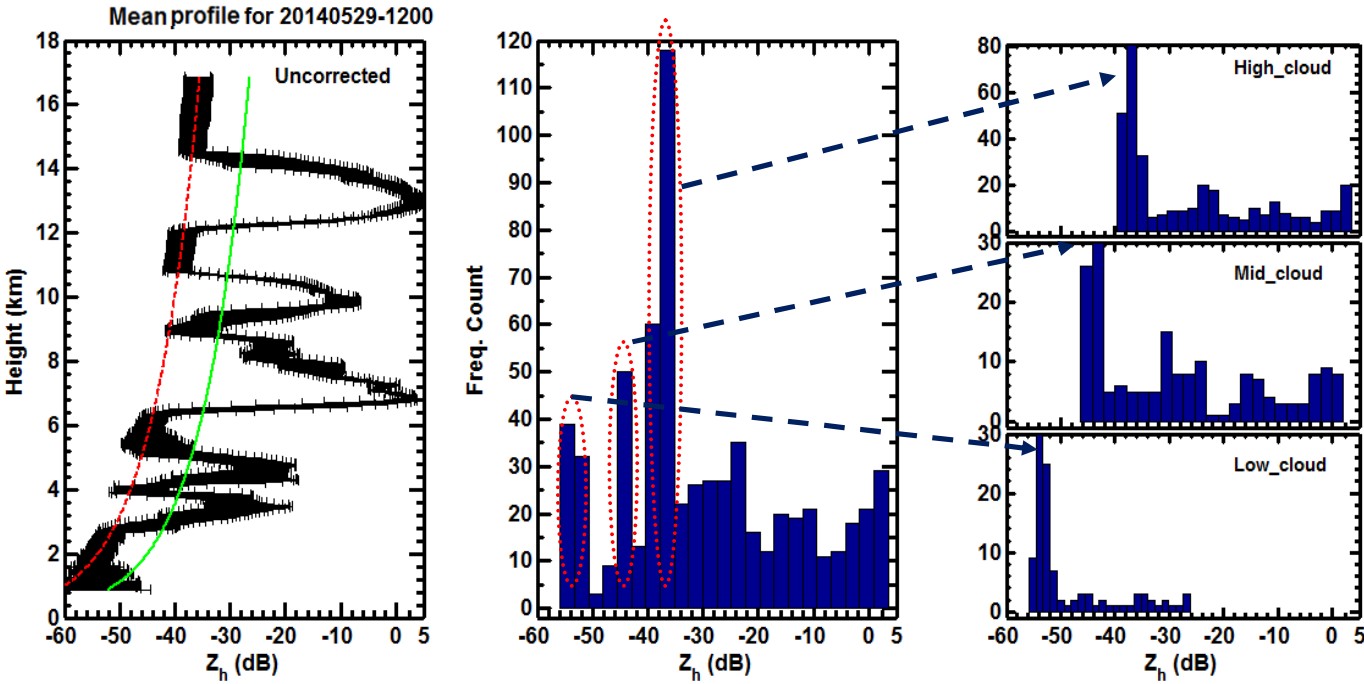

**Figure 9a: (Left panel)** Uncorrected mean reflectivity profile on 29 May 2014 during 1200-1205 UT superimposed with curves S1 (dashed red line) and S5 (solid green line). Histogram of Z profile (Middle panel). (left three sub panels) for altitude regions of low (<3.6 km), mid (3.6 km>=ht<8.6 km) and high (>=8.6 km). The right sub panels each peak of histogram are mapped on to the corresponding three peaks with the whole vertical structure of Z. This infers the noise clearly suppresses the meteorological information.

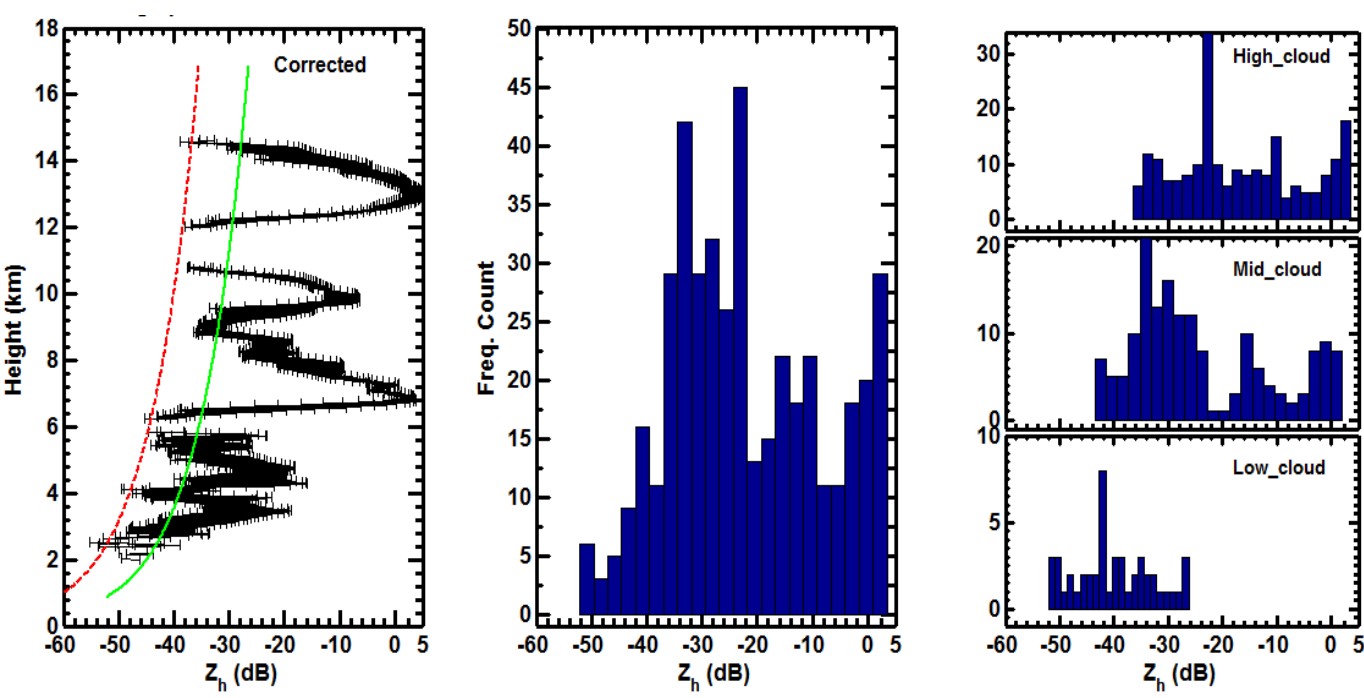

**Figure 9b:** Same as 9a but it is corrected by filtering out noise and biota. The correction applied to Z profile allows to pop-up the true meteorological cloud reflectivity distribution.

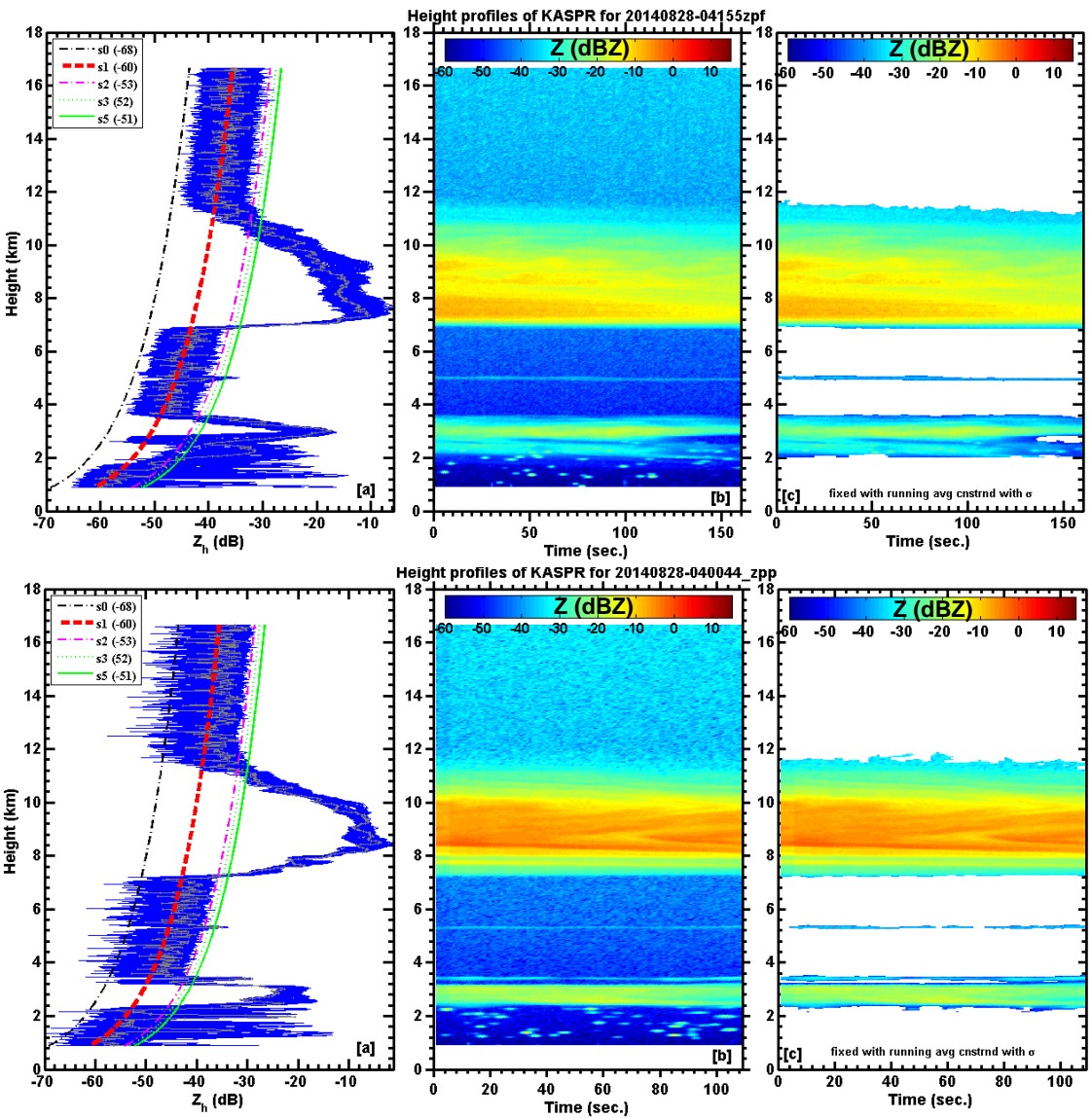

**Figure 10: Same as 7 but for vertical looking KaSPR measurements at 0400 UT on 28 Aug 2014 using (top) FFT processing (bottom) 15 minutes prior one using PP processing. PP case will be used further to evaluate the polarimetric algorithm performance.**

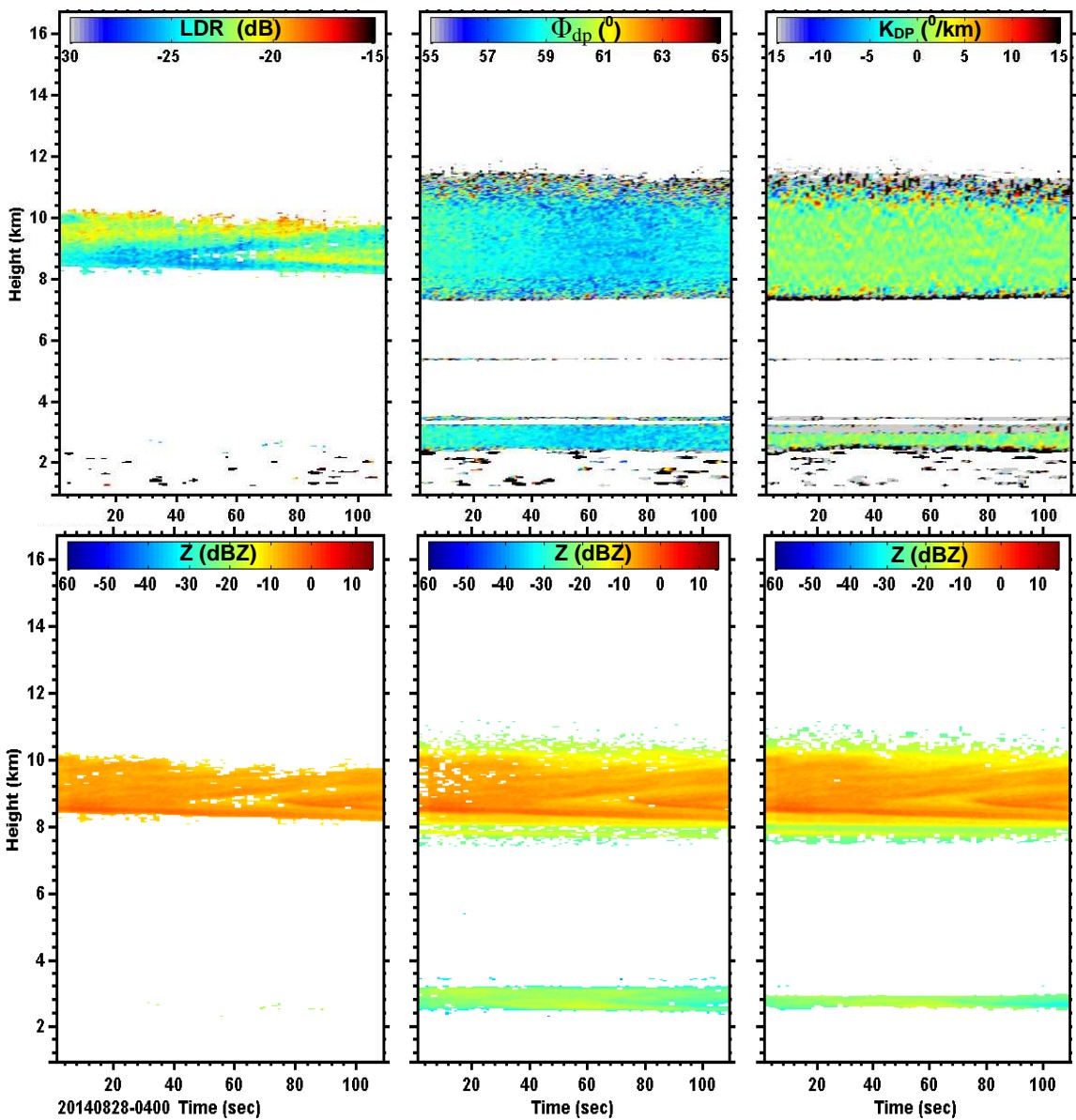

**Figure 11: HTI plots of (top panel) LDR, $\Phi_{dp}$ and $K_{DP}$ parameters pertinent to PP processed data of Figure 10 and (bottom panels) biota filtered reflectivity after applying corresponding polarimetric thresholds of the respective top panels.**

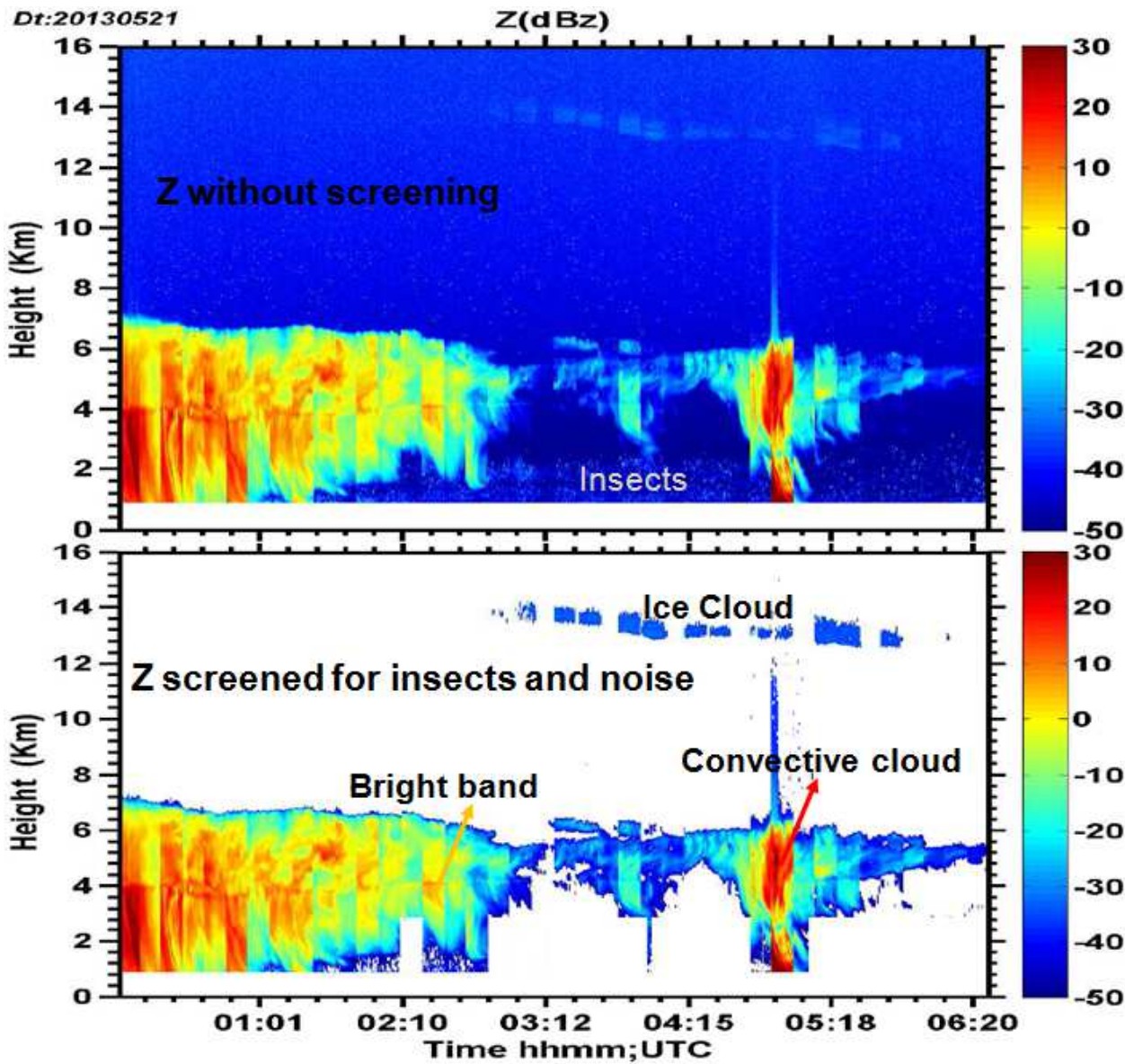

**Figure 12a: (Top) Same as Figure 7b (uncorrected) and (bottom) same as Figure 7c (corrected) but integrated for duration of 0000-0630 UT taken at an interval of ~ 15 minutes on 21 May 2013**

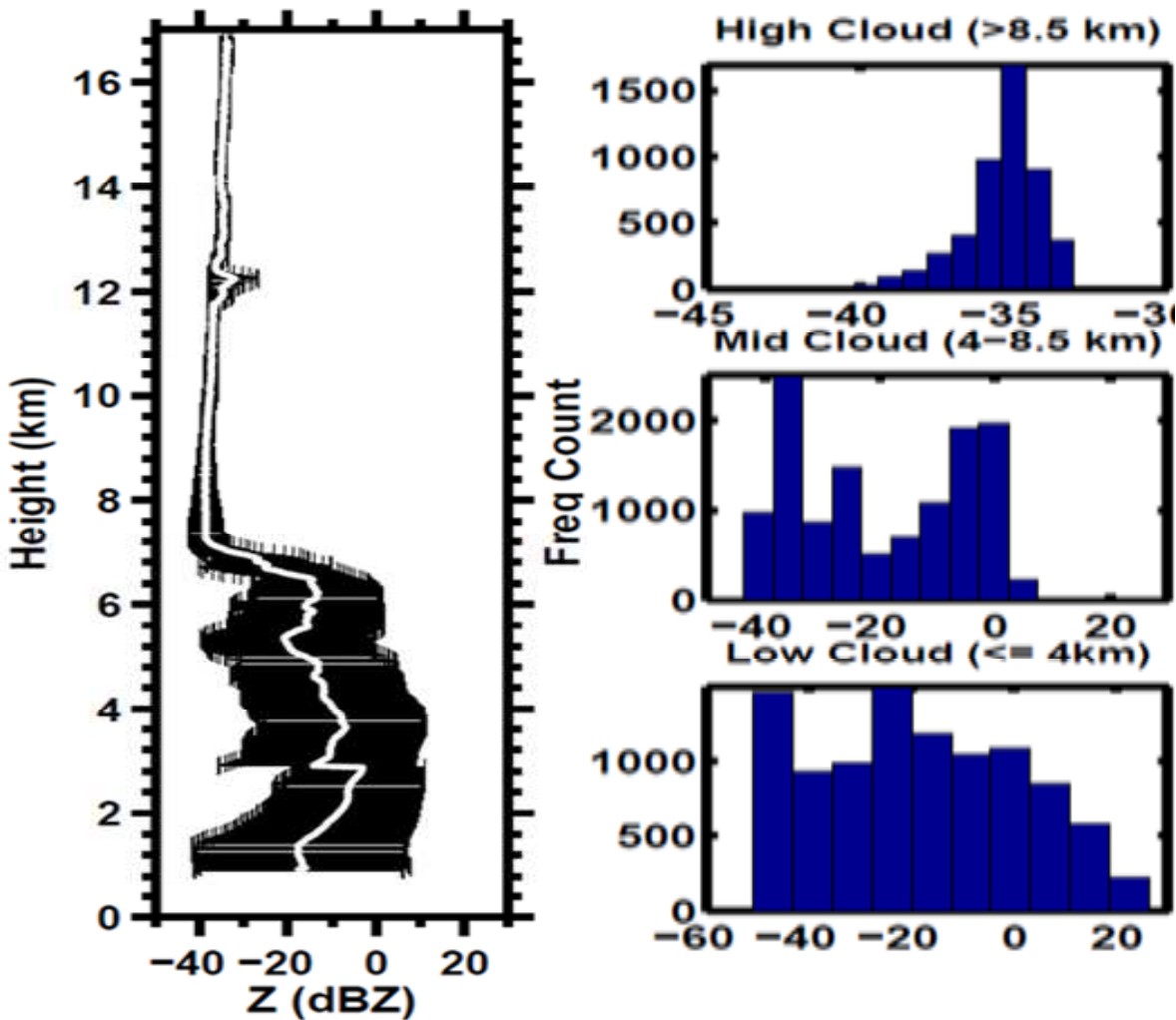

**Figure 12b:** Screened-out cloud radar reflectivity mean and standard deviation profile with the tri-model cloud reflectivity frequency distribution.

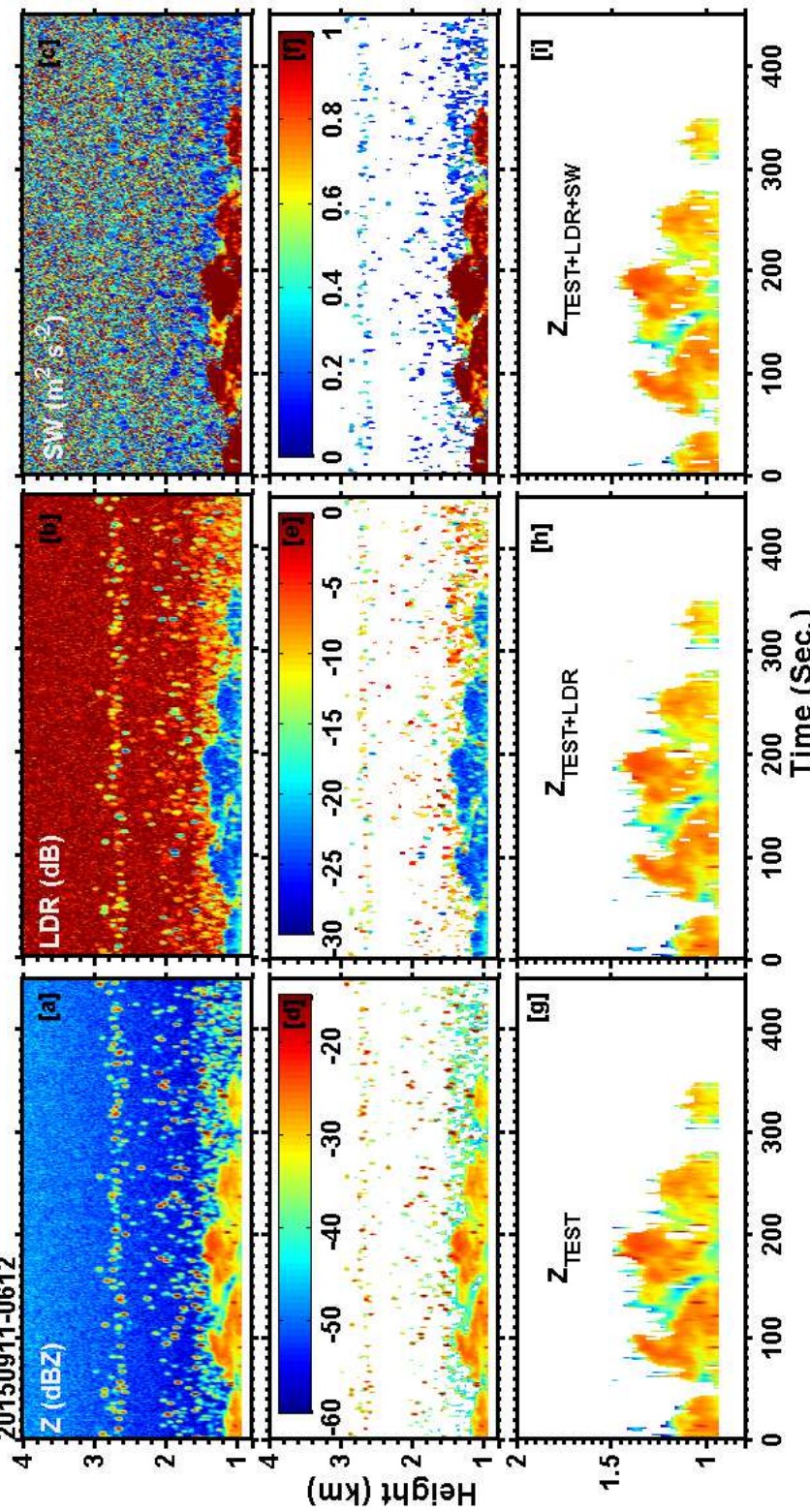

**Figure 13: Cloud radar measurements of reflectivity (Z), LDR, Spectral Width (SW) with noise (a-c) and filtered out for noise using S5 curve (d-f), TEST algorithm screened output Z for clouds (g), g + biota filtering using LDR > -14 dB (h), h + SW filter for biota using SW < 0.5 m²s⁻² (i).**

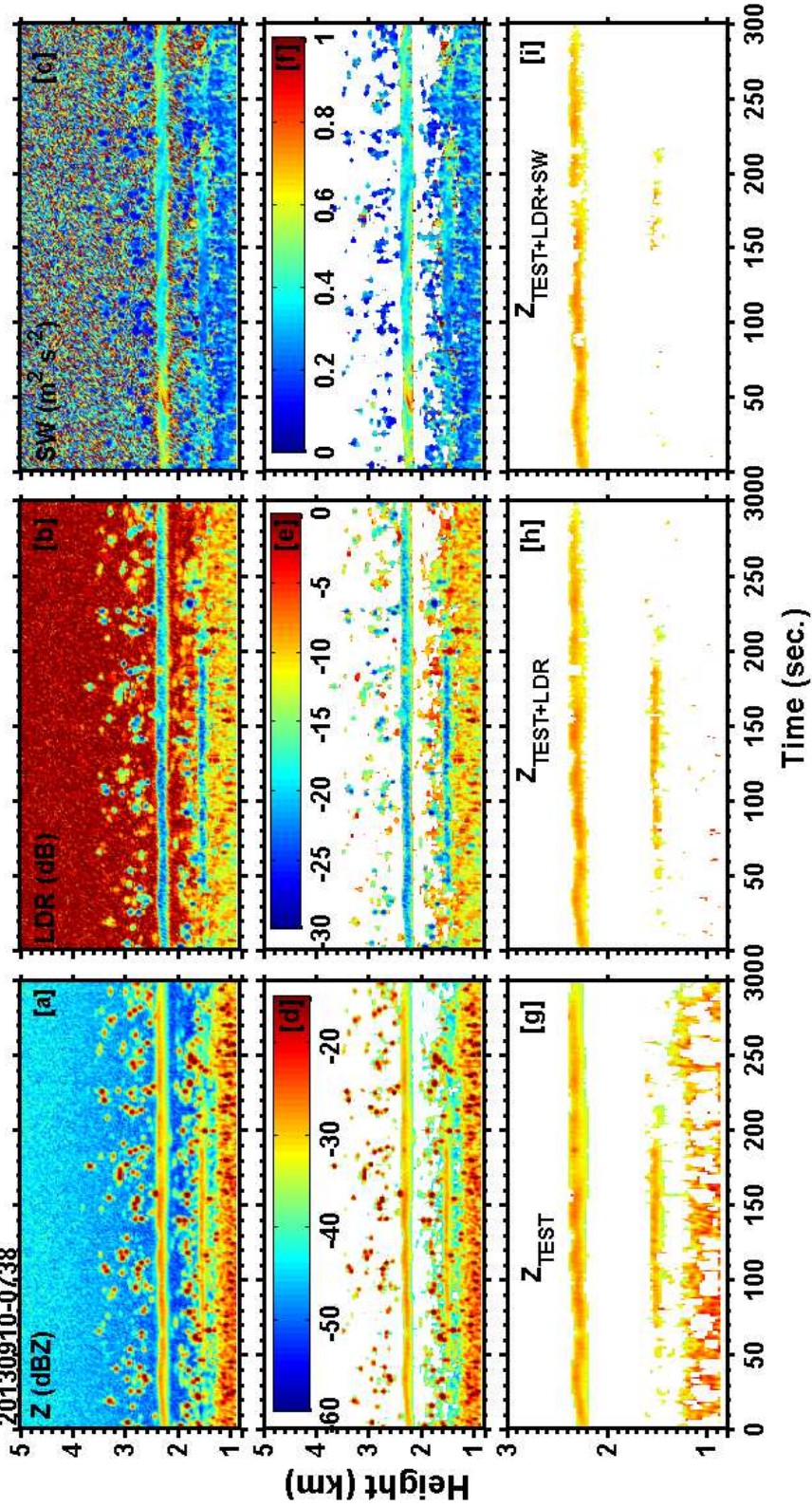

**Figure 14: Same as figure 13 but for typical high density b noted during 0738 UT on 10 Sep. 2013.**

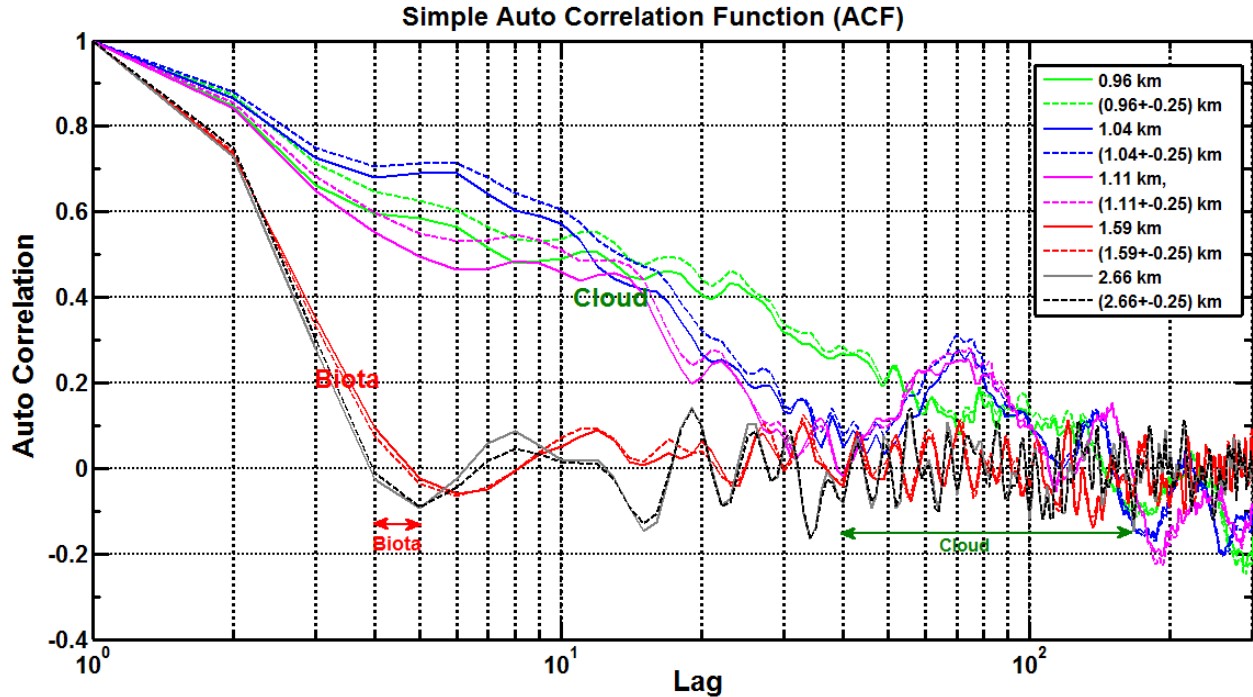

**Figure 15:** Simple ACF inferred de-correlation periods associated with shallow cumulus cloud (base, mid and top) and biota height levels with the reflectivity measurements of figure 13a.

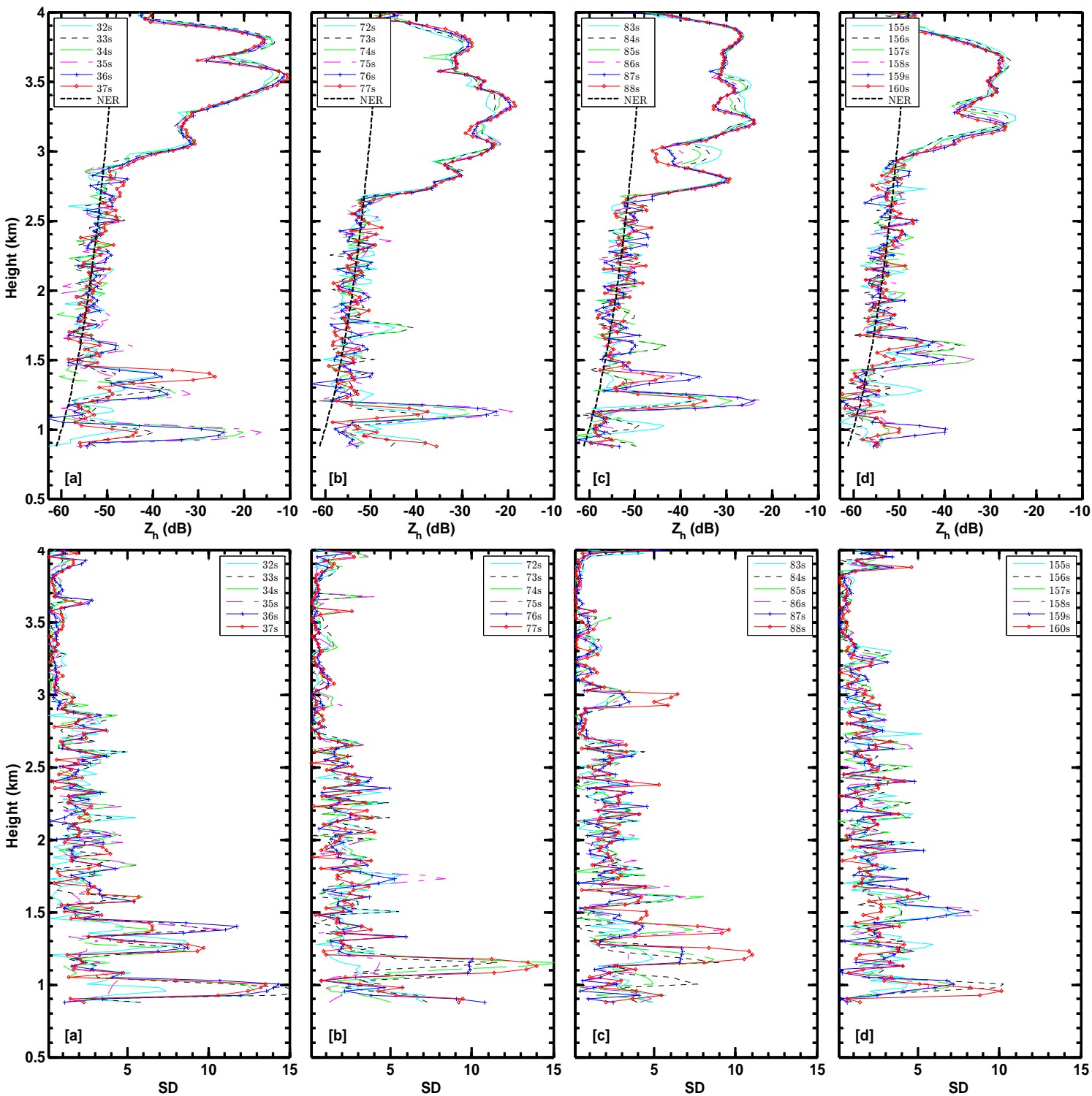

**Figure A1: Instentaneous height profiles of Z during 1200-1205 UT on 29 May 2014 with centered numer profile notice to be the strong biota return identified with HTI plot of figure 4b. Bottom panesl correspond to standard devation (SD) from four point running average.**

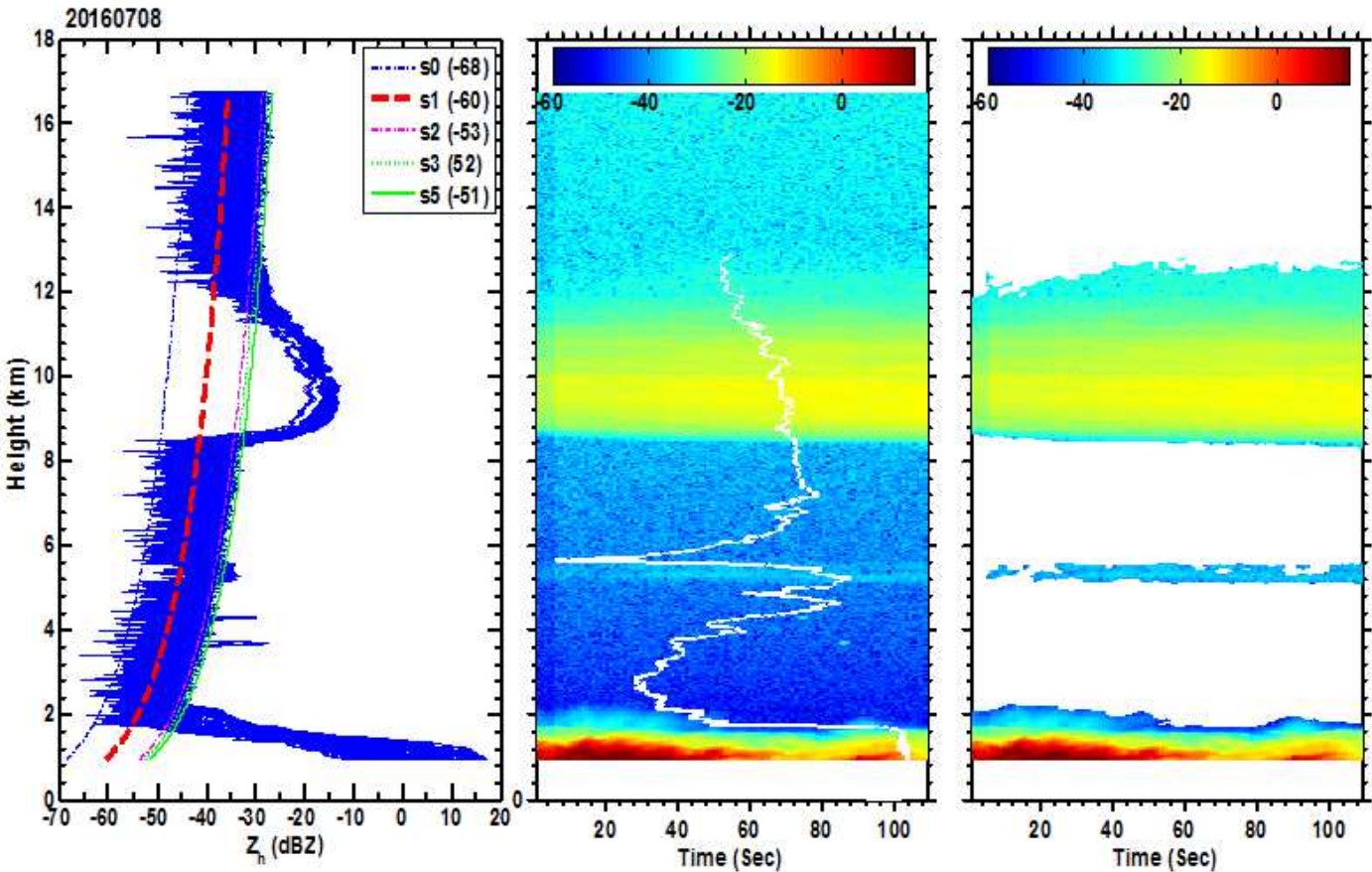

**Figure A2: (Right-middle-left) Same as 1(a-c) but on 08 Jul 2016 during 0531 UT for the duration of 108 sec. S0-S5 are NER curves. Collocated GPS-RS relative humidity (%) profile had shown as while solid line in the middle panel.**

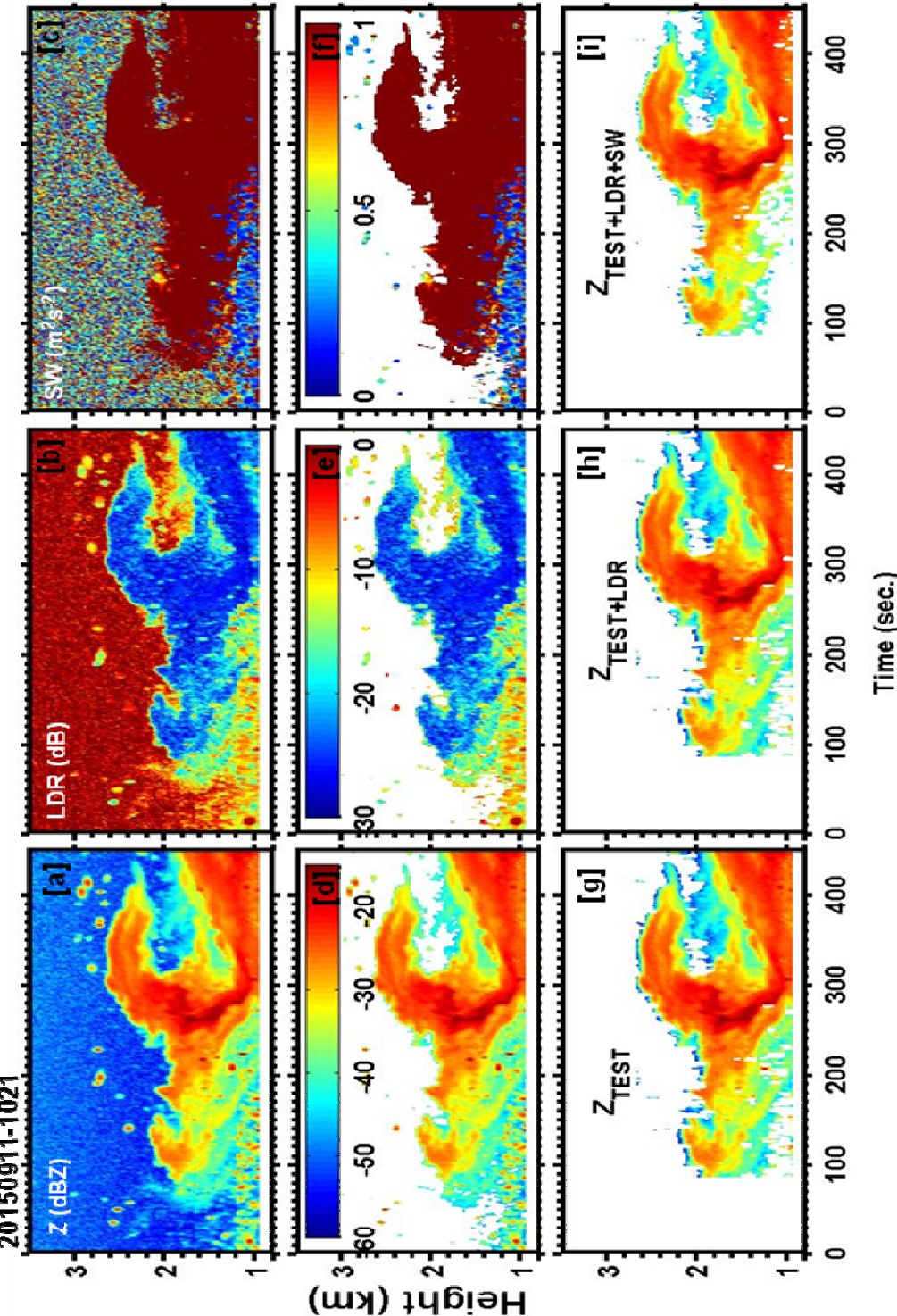

**Figure A3:** Same as figure 13 but during 1021 UT on 11 Sep. 2015 for the duration of 449 sec.