# Peer review of "A simple biota removal algorithm for 35-GHz cloud radar"

_Atmospheric Measurement Techniques, 2017_

## Referee Comment (RC1) · Anonymous Referee #1 · 28 Sep 2017

Reviewer comments on the manuscript titled "simple insect removal algorithm for 35-GHz cloud radar measurements", by Madhu Chandra R. Kalapureddy et al.

The study presented a technique which uses high temporal and spatially resolved reflectivity profiles to extract the cloud echoes from the clutter (mainly from the biota). The proposed technique suggested as a simple and efficient solution for clutter removal, compared to earlier sophisticated techniques based on dual polarization and spectral techniques. I think manuscript has several shortcomings, related to technique and assumptions, poor job of literature review, references and lack of solid conclusions. In its entirety, I would recommend rejection of this paper in its present form.

Major comments:

[Figure]

The screening technique authors have implemented using simple measures of reflectivity (or SNR) thresholds and its variability to filter out the clutter has been a usual practice in the cloud radar community as a part of post-processing exercise. The challenge of separating insects from the cloud clutter is difficult due to the lack of clear demarcation between their properties as seen by cloud radar. More often than otherwise, the screening process requires more than one variable, which captures the texture, distribution width, and physical properties of these echoes. With this motivation, some of the earlier studies have devoted their efforts to address this problem using different techniques (fuzzy-logic, spectral technique or polarization properties).

The authors haven't clearly appreciated and addressed the insect removal to the detail that it was needed. They have demonstrated the algorithm with several minutes of data, which doesn't warrant the techniques robustness to apply for other conditions. Authors have made several assumptions about the insect layer depth, their decorrelation timescale without presenting any evidence about the location of the shallow boundary layer clouds, where the insect clutter is very critical. Previous studies (e.g., Geerts and Miao 2005; Chandra et al., 2010) have utilized the long-term observations of insect echoes to study the convective boundary layer, where they have shown that the insect decorrelation times may vary from few seconds to few minutes depends on boundary layer organization. The authors would have shown the distribution of the cloud base locations (from the closest ceilometer data) to justify their presumed insect layers below ~2km. I suggest authors to utilize the supplemental observations (such as ceilometer, microwave radiometer) to present the cloud properties and refine their insect-cloud algorithm based on the locations of cloud layer depth.

As an alternative solution to the computationally intensive spectral techniques for the insect clean-up (e.g., Luke et al., 2008), a computationally efficient technique to minimize insect clutter have been implemented based on fuzzy-logic algorithm (e.g., Chandra et al., 2013), which takes into account both the physical properties of clouds and different radar moments. This technique can be applied with different levels of complexity based on the supplemental observations (Microwave Radiometer/Ceilometer) you have in addition to radar moments. I suggest authors go through this technique for more details.

The basis of the present technique is that the reflectivity distribution could be effective in separating insects from clouds, which may not be the case always. There could be instances when the range of reflectivities from the shallow passive clouds could be similar to the insects (refer to panels, a1 and a2 from the Fig. 13 as in Chandra et al., 2013). This study has taken into account not only the physical properties of cloud (e.g., liquid water path) but also texture signatures in the reflectivity field, the variability of the scatterers inside the radar range resolution from the spectrum width variable-one of the main predictors in insect-cloud separation.

The authors would have shown the technique demonstration effectively with few figures. I feel that there are some figures (Figure 8a and 8b, Figure 11) which don't serve any purpose. Some of the references (cited in the lines 64-98) related to the clutter removing techniques implemented at other frequencies (C- S-Band) were not necessary.

---

## Referee Comment (RC2) · Anonymous Referee #2 · 5 Oct 2017

I think the NER algorithm should be removed from this paper. There are much more general ways for thresholding between signal and "salt and pepper". I should be done by the radar software. so that it adapts automatically to the processing parameters.

I think the TEST algorithm for filtering insect echoes from the radar data is helpful if it is used in combination with LDR-filtering and or dual frequency filtering. The author comes to this conclusion in the lines around 285 and I agree to it. In the rest of the paper the algorithm is described as a standalone alternative to LDR or dual-frequency filtering. It should be clearly said that this does not work as in regions with much insects the insect signatures are as smooth as butter. There they are volume filling targets.

The theoretical background of the algorithm should be explained more general:

[Figure]

actually the signal from insects has a longer de-correlation time than signal from volume filling targets.

The signal from volume targets is a sum of many signals with statistical phases and amplitudes which causes noise with normal distribution (central limit theorem). Therefore even if the volume is filled with stationary targets (droplets falling with different speeds, some exiting the volume, some entering) each line of of the un-averaged complex spectra is normal distributed noise with zero mean and a variance corresponding to the power in the doppler spectrum. The doppler spectra are the abs-square of the complex spectra and therefore they are still noisy. Due to squaring the distribution is transformed from normal to exponential. After averaging over 1 s this noisiness has smoothed out by 1/sqrt(nave). In contrast the signal from a single insect is not noisy at all if its SNR is large. But there is another reason causing variance in the biota signals. Typically the insects are advected through the radar beam,entering with apparent downward velocity and leaving with apparent upward velocity. The pass through time depends on beam width (deg), height, and wind speed. this causes a spiky spectra. if there are not too many insects, then there is a maximum in the variance spectrum of biota signals at 1/(pass through time). For this reason the variance spectrum of volume targets is white and for biota with moderate densities it has a maximum at the frequency corresponding to the 1/(pass through time). The TEST-procedure extracts the variance caused by biota by cancelling the high frequency variance of the volume targets by 1 second averaging and by cancelling the low frequency variance with high pass filtering the variance of reflectivities. The remaining medium frequency componets of the variance spectrum is dominated by the beam passing of the single insects, and therefore it can be used for recognising if the signal is from biota or clouds. Without understanding the author found that the test method works in many cases. In cases with too high or too low wind speed this simplified filtering may fail.

Here are some minor notes:

45: sensible —> sensitive

56: to our experience the reflectivities of biota are below 0 dBZ, reflectivities of rain are above 0 or 5 dBZ.

95: T-matrics —> Rayleigh

96: I would change the sequence from large to small 1 droplets with .1 mm : -60 dBZ 64 droplets with 0.05: -60 dBZ 1e6 droplets with 0.01: -60 dBZ

I guess the author wants to say that hydro meteors are volume filling targets in most cases. For a single spectral component or say a single drop D size $Z = N\ D^6/V$, where V is the radar volume which about 1000 to 25 000 m^2 depending on height, and N is the number of droplets in the radar volume. In case of single target N=1 and therefore $Z\_singletarget = D^6/V$ or $Z\_volumetraget \gg D^6/V$. As D can be inferred from the terminal falling velocity which is roughly the doppler velocity at least for larger droplets, it can be found by analysing data that hydrometeors are volume filling in the the majority of cases. Sometimes large droplets in the beginning of a rain event are rather single targets.

98: is the PRF of this radar really adjusted to such a low value. this would allow for a maximum range of 300 km which is not useful in vertically pointing mode. a prf of 7 to 10 khz is more adequate in vertical mode. this allows a much larger velocity range. but this is not relevant for the scope or this paper.

I cannot understand or even guess the mening of this sentence.

126: ..more than 2 m/s and the de-correlation

137: This method will be fully explained in the following section —> It seems it is in the rest of this section and then in the section Results and discussion beginning in line 214

139: fixing —> thresholding

227: This is not true for cyrus clouds. The have a very soft top.

---

## Referee Comment (RC3) · Anonymous Referee #3 · 5 Oct 2017

General comments and recommendation for disposition of the paper

Millimeter-band radars are very sensitive to detect small targets such as cloud droplets and also insects and other biological particulates (biota) present in great number in the lower atmosphere. Polarization measurement is an efficient mean to discriminate cloud echoes from non meteorological scatterers that share in common very low reflectivity. Unfortunately most radars are not equipped with polarization measurements. This short paper proposes for these standard radars a simple technique able to separate meteorological and non-meteorological echoes. It uses only successive vertical reflectivity profiles acquired by a 35-GHz radar operated at vertical incidence with a 50 m pulse length and one second temporal sampling. Because of the high spatial and temporal resolution, most of the time only one or no biota target is present in the

pulse resolution volume. In contrast, cloud echo is due to millions droplets that fulfill the pulse volume. As a consequence signal variability at a given range between two vertical profiles is much more important for biota scatterers than for cloud echoes. Signal variability is given here by the standard deviation of the reflectivity over the time of five profiles that corresponds to the typical duration of the biota echoes crossing the antenna beam. The threshold value that separates distinctly biota from cloud is obtained from statistical analysis of a large radar observation set. Indeed this value should be adjusted for a radar having different characteristics. The topic of this study enters the scope of the journal and responds to a real issue for anybody who wants to extract physical quantities from radar signal. The work is put into perspective with past equivalent investigations through a large panel of bibliographic references. The work based on well chosen graphics is convincing and above all the methodology is validated with polarization measurements provided by the same radar. In conclusion this paper that presents a good scientific interest is suitable for publication in Atmospheric Measurement Techniques Journal. However this recommendation is subordinated to the authors consideration of the following comments.

Main review points

1) Lines 48 to 50 give the list of the source of non-meteorological echoes which comprises insects and other biological particulates (biota). The title refers only to insect and in the text the word insect is nearly always used. Even if the insects is the main source of biological echoes it is a restrictive term. I propose to use in place the word biota introduced by the authors.

2)- In figure A2 strong vertical gradient of humidity is associated with the presence of cloud echoes. We may deduce that also strong refractivity index gradient exists which can be a potential source of Bragg or specular echoes. For information an explanation that this type of echoes, observable with UHF and VHF band radars, has a very low probability to be detected by millimeter-band radars will be welcome.
3)- The sensitivity of the radar is -60 dBZ at 1 km range (line 95). This value seems to me very optimistic according to the radar characteristics. Give some details on the radar calibration.

4)- May be the high radar sensitivity is due to the use of pulse compression (Table 1). If this mode is used give the effective pulse length, the code moments number and the lower range gate available for the data set presented in the paper.

5)- The term point target is used line 102 for non-meteorological echo. In fact a scatterer is named punctual echo when it is alone in the pulse volume. In that case echo duration is related to the time taken by the target to cross the radar beam, to its radar cross-section and its position relative to the beam axis. All these factors explain the signal variability of biota echoes.

6)- In fig.1, and others equivalent figures, a range (r) correction of the radar signal of the form r2 is used (line 109). It is correct for volume echoes such as cloud echoes, for point targets it is inadequate. The range correction for such backscatters has the form r4.

7)- When there is an echo at a certain range, the signal at the receiver output is the sum of the receiver noise voltage and the detected backscattered wave. It is therefore necessary to remove the noise power in order to get the backscattered power. It is evident that this has not been done for the presentation showed in the figures such as fig.1.

8)- Line 111: Receiver noise is made of thermal noise generated within the receiver chain and also of other sources which are taken into account through the noise figure of Table 1.

9)- Give more details on the computation of the running mean and standard deviation (line 136) of the successive vertical profiles of reflectivity. In particular it is important to precise if these quantities are computed before or after noise removal.

10)- Line 161: Receiver noise is not en echo but a signal generated in the receiver chain.

11)- A statistical de-correlation time is introduced line 174. I do not understand very well how it is computed. I think it is related to the standard deviation of the reflectivity. Give the formula that links de-correlation time and reflectivity standard deviation. In figures 3 and in the text the unity used for the standard deviation is not given.

12)- lines 218 to 219 ...biota that are found to extend less than 2-4 height bins each of 25 m... : vertical spreading of a point echo is expected to extend over half pulse. How do you explain this large spreading that can approach 2 pulse lengths. Is the use of a compressed pulse that produces this increase.

13)- I suppose that the radar has Doppler capability because line 263 and 264 Pulse-Pair and Fourier Transform are cited. Doppler spectra width contains information at the pulse scale on the de-correlation time of the echoes. It could have been used instead of the reflectivity standard deviation. Did you try to analyze this quantity to discriminate echo type.

---

## Author Comment (AC1) · 26 Oct 2017

Effectiveness of TEST algorithm lies in the simultaneous presence of shallow ABL cloud and biota at the same height level which was missed to address clearly in the submitted Manuscript. Here we would like to fill that gap for much clarity.

TEST algorithm is able to filter out biota and preserve the shallow ABL layer clouds (figure AC1(c)) when both the echoes are possessing same range of reflectivity values. This further shows superior biota removal capability of TEST. Smaller de-correlation period associated with biota are further confirmed with less spectral width values (<0.3 m2 s-2; figure AC1(d)). Higher spectral width values, of the order of ∼ 1 m2 s-2 of the cloud indicates the random motion of the smaller particles of cloud with in radar

scattering volume are affected by the ABL turbulence. Shallow ABL cloud regions show LDR values <-20 dB whereas insects shows varied LDR values in the range of -25-to -5. Thus, LDR alone is not sufficient to remove all insects. Adding TEST used shorter running average period of 4-5 second with LDR enables one to filterer out non-meteorological contribution from the radar returns. It is interesting to note that TEST algorithm with SW could be future efficient combination to filter all biota once we ensure that first few range bins are not affected by the local clutter.

Please also note the supplement to this comment:
https://www.atmos-meas-tech-discuss.net/amt-2017-254/amt-2017-254-AC1-supplement.pdf
* * *
[Figure]

**Fig. 1.** Figure A1: HTI plot of cloud radar measured (a) Reflectivity (Z), (b) noise removed Z, (c) TEST filtered Z, (d) Spectral Width (SW), and (e) LDR at 0612 UT on 11 Sep 2015.

---

## Author Comment (AC2) · 24 Nov 2017

We are thankful to the Anonymous Referee 1, for the hearty services in rendering his/her experience and knowledge based comments, those are valuable to us for improving the quality and the focus of the paper. The point-to-point AR1 responses of the authors are as below:

AR1-Comment: The study presented a technique which uses high temporal and spatially resolved reflectivity profiles to extract the cloud echoes from the clutter (mainly from the biota). The proposed technique suggested as a simple and efficient solution for clutter removal, compared to earlier sophisticated techniques based on dual polarization and spectral techniques. I think manuscript has several shortcomings, related to

[Figure]

technique and assumptions, poor job of literature review, references and lack of solid conclusions. In its entirety, I would recommend rejection of this paper in its present form.

Response: Thank you! We request AR1 now to review the latest modified version of the paper where we re-written the whole introduction part the manuscript (MS) and cited all possible concern references that come under the scope of the paper and responded also to other valuable referee points. Furthermore, clarity on the technique/algorithm and its main application region has now been clearly come through the revision process in the current modified version of the MS at first Para of section 2 (pg 3) and added basis for TEST (Line 292-295, pg 4-5), including new figures (fig 13 to fig 15) show the potential of TEST in screening out clouds by filtering out biota. Further weakness of TEST under challenging conditions like within cloud and high density biota has been overcome using extra measurements like LDR and SW. This can be seen at the last two paragraphs of Results and Discussion. Thus, the main conclusion of the paper is how simplest way one can remove the biota contribution and preserve true cloud hydrometeor echo and its need for the study of important shallow cumulus/ABL clouds before the actual cloud radar echo weighted measurements consider for any research application purpose. The above revision asked necessary modification to the last section (Summary and Conclusions) from page 11 onward.

AR1-Major comments: The screening technique authors have implemented using simple measures of reflectivity (or SNR) thresholds and its variability to filter out the clutter has been a usual practice in the cloud radar community as a part of post-processing exercise. The challenge of separating insects from the cloud clutter is difficult due to the lack of clear demarcation between their properties as seen by cloud radar. More often than otherwise, the screening process requires more than one variable, which captures the texture, distribution width, and physical properties of these echoes. With this motivation, some of the earlier studies have devoted their efforts to address this problem using different techniques (fuzzy-logic, spectral technique or polarization properties).

Response: Separating biota from cloud is difficult and challenging but not impossible by the cloud radar if one effectively makes use of advancing radar and signal processing technique (e.g., chirping and DSP) that enables to have the provision of high spatial and temporal resolution radar measurements (1st paragraph of System, Data and Methodology) that can demarcate the cloud echo from insects for example through reflectivity texture (e.g., TEST). With our knowledge, TEST, it is first of its kind effort to consider both reflectivity variance (i.e., dBZ texture) and its rate of change through running average for every 4 seconds. The above point pragmatically working as to identify the time coherence or de-correlation periods associated with clouds and biota echo signature (see newly added figure 15 and its description at pg 11). Moreover, the de-correlation can be evidenced through direct third Doppler power spectral moment; spectral width measurements that clearly show biota exhibits less velocity variance thus the relatively quicker time de-correlation at the pulse scale. In Fig. 2, the zoomed portion of Fig 1, the rounded echo confirms the presence of non-hydrometeor information by their duration of maximum 10 sec which is too small for a cloud to form and then suddenly disappear. So the vertical extension and the time duration of the echoes are the two key factors to discriminate cloud from non-meteorological information. Merits and de-merits of TEST has been brought out exclusively with Figure 13 and 14 that are making use of LDR and Spectral width measurements besides to Z to enhance the proposed TEST algorithm capability under tough conditions like cloud under heavily dense insects clutter.

AR1-Comment: The authors haven't clearly appreciated and addressed the insect removal to the detail that it was needed. They have demonstrated the algorithm with several minutes of data, which doesn't warrant the techniques robustness to apply for other conditions. Authors have made several assumptions about the insect layer depth, their decorrelation timescale without presenting any evidence about the location of the shallow boundary layer clouds, where the insect clutter is very critical. Previous studies (e.g., Geerts and Miao 2005; Chandra et al., 2010) have utilized the long-term observations of insect echoes to study the convective boundary layer, where they have shown

that the insect decorrelation times may vary from few seconds to few minutes depends on boundary layer organization. The authors would have shown the distribution of the cloud base locations (from the closest ceilometer data) to justify their presumed insect layers below âĹij2km. I suggest authors to utilize the supplemental observations (such as ceilometer, microwave radiometer) to present the cloud properties and refine their insect-cloud algorithm based on the locations of cloud layer depth.

Response: Unfortunately the suggested useful complemented data was not available at and around the radar site. However using some available GPS RS observations from the radar site, the presence of weaker clouds have been proved with auxiliary Figure A2 (please also see the response to the comment 2 of AR#3). Further, we consider the reviewer's well suggested point on the inclusion of shallow boundary layer cloud case with insects clutter when both have near same reflectivity values (added Figures 13-14). In fact, thanks to the reviewer that now it is clearly illustrating the potential of TEST that lies mostly to the ABL, where shallow cloud evolves, where the affinity of biota are predominant. Below are two examples of such low level/ shallow cumulus clouds with biota clutter where the fine performance of TEST is evident.

Note: Referee figure quoted as 'Figure ARX' and MS figure as 'Figure X' Please also see pdf responses attached

Please also note the supplement to this comment:
https://www.atmos-meas-tech-discuss.net/amt-2017-254/amt-2017-254-AC2-supplement.pdf

[Figure]

[Figure]

**Fig. 1.** Figure AR1: HTI plot of cloud radar measured (a) Reflectivity (Z), (b) noise removed Z, (c) TEST filtered Z, (d) Spectral Width (SW), and (e) LDR at 0612 UT on 11 Sep 2015.

[Figure]

**Fig. 2.** Figure AR2: TEST performance in filtering biota echoes that are co-present with low clouds.

**Supplement:**

**Authors Responses**

To the *Interactive comments on* manuscript titled "Simple insect removal algorithm for 35- GHz cloud radar measurements", M C R  Kalapureddy et al. *Atmos. Meas. Tech. Discuss., doi:10.5194/amt-2017-254, 2017*

**At the outset, we are grateful to the Editor(s) and all Editorial team for their services/help and untiring timely support and cooperation. We are also equally thankful to all the three Anonymous Referees, for their hearty services in rendering experience and knowledge based comments, those are valuable to us for improving the quality and the focus of the paper.**

**The point-to-point AR1 responses of the authors are as below:**

**Anonymous Referee #1** (AR1)

AR1-Comment: The study presented a technique which uses high temporal and spatially resolved reflectivity profiles to extract the cloud echoes from the clutter (mainly from the biota). The proposed technique suggested as a simple and efficient solution for clutter removal, compared to earlier sophisticated techniques based on dual polarization and spectral techniques. I think manuscript has several shortcomings, related to technique and assumptions, poor job of literature review, references and lack of solid conclusions. In its entirety, I would recommend rejection of this paper in its present form.

**Response:  Thank you! We request AR1 now to review the latest modified version of the paper where we re-written the whole introduction part the manuscript (MS) and cited all possible concern references that come under the scope of the paper and responded also to other valuable referee points.  Furthermore, clarity on the technique/algorithm and its main application region has now been clearly come through the revision process in the current modified version of the MS at first Para of section 2 (pg 3) and added basis for TEST (Line 292-295, pg 4-5), including new figures (fig 13 to fig 15) show the potential of TEST in screening out clouds by filtering out biota. Further weakness of TEST under challenging conditions like within cloud and high density biota has been overcome using extra measurements like LDR and SW. This can be seen at the last two paragraphs of Results and Discussion.  Thus, the main conclusion of the paper is how simplest way one can remove the biota contribution and preserve true cloud hydrometeor echo and its need for the study of important shallow cumulus/ABL clouds before the actual cloud radar echo weighted measurements consider for any research application purpose. The above revision asked necessary modification to the last section (Summary and Conclusions) from page 11 onwards.**

AR1-Major comments: The screening technique authors have implemented using simple measures of reflectivity (or SNR) thresholds and its variability to filter out the clutter has been a

usual practice in the cloud radar community as a part of post-processing exercise. The challenge of separating insects from the cloud clutter is difficult due to the lack of clear demarcation between their properties as seen by cloud radar. More often than otherwise, the screening process requires more than one variable, which captures the texture, distribution width, and physical properties of these echoes. With this motivation, some of the earlier studies have devoted their efforts to address this problem using different techniques (fuzzy-logic, spectral technique or polarization properties).

**Response: Separating biota from cloud is difficult and challenging but not impossible by the cloud radar if one effectively makes use of advancing radar and signal processing technique (e.g., chirping and DSP) that enables to have the provision of high spatial and temporal resolution radar measurements (1st paragraph of System, Data and Methodology) that can demarcate the cloud echo from insects for example through reflectivity texture (e.g., TEST). With our knowledge, TEST, it is first of its kind effort to consider both reflectivity variance (i.e., dBZ texture) and its rate of change through running average for every 4 seconds. The above point pragmatically working as to identify the time coherence or de-correlation periods associated with clouds and biota echo signature (see newly added figure 15 and its description at pg 11). Moreover, the de-correlation can be evidenced through direct third Doppler power spectral moment; spectral width measurements that clearly show biota exhibits less velocity variance thus the relatively quicker time de-correlation at the pulse scale. In Fig. 2, the zoomed portion of Fig 1, the rounded echo confirms the presence of non-hydrometeor information by their duration of maximum 10 sec which is too small for a cloud to form and then suddenly disappear. So the vertical extension and the time duration of the echoes are the two key factors to discriminate cloud from non-meteorological information. Merits and de-merits of TEST has been brought out exclusively with Figure 13 and 14 that are making use of LDR and Spectral width measurements besides to Z to enhance the proposed TEST algorithm capability under tough conditions like cloud under heavily dense insects clutter.**

AR1-Comment: The authors haven't clearly appreciated and addressed the insect removal to the detail that it was needed. They have demonstrated the algorithm with several minutes of data, which doesn't warrant the techniques robustness to apply for other conditions. Authors have made several assumptions about the insect layer depth, their decorrelation timescale without presenting any evidence about the location of the shallow boundary layer clouds, where the insect clutter is very critical. Previous studies (e.g., Geerts and Miao 2005; Chandra et al., 2010) have utilized the long-term observations of insect echoes to study the convective boundary layer, where they have shown that the insect decorrelation times may vary from few seconds to few minutes depends on boundary layer organization. The authors would have shown the distribution of the cloud base locations (from the closest ceilometer data) to justify their presumed insect layers below ~2km. I suggest authors to utilize the supplemental observations (such as ceilometer, microwave radiometer) to present the cloud properties and refine their insect-cloud algorithm based on the locations of cloud layer depth.

**Response: Unfortunately the suggested useful complemented data was not available at and around the radar site. However using some available GPS RS observations from the radar site, the presence of weaker clouds have been proved with auxiliary Figure A2 (please also see the response to the comment 2 of AR#3). Further, we consider the reviewer's well suggested point on the inclusion of shallow boundary layer cloud case with insects clutter (Figure AR1 and AR2) when both have near same reflectivity values (added Figures 13-14). In fact, thanks to the reviewer that now it is clearly illustrating the potential of TEST that lies mostly to the ABL, where shallow cloud evolves, where the affinity of biota are predominant. Below are two examples of such low level/ shallow cumulus clouds with biota clutter where the fine performance of TEST is evident.**

[Figure]

**Figure AR1:** HTI plot of cloud radar measured (a) Reflectivity (Z), (b) noise removed Z, (c) TEST filtered Z, (d) Spectral Width (SW), and (e) LDR at 0612 UT on 11 Sep 2015.

For demonstration, few typical cases of several minutes have been presented at the beginning but for robustness and application of this algorithm as suggested has been demonstrated with Figure 12 that makes use of several contiguous vertical looking measurements files in a day for more than 6 hours duration. In fact, we are thoroughly using this algorithm for all our cloud radar data (2013-2016) for quality cloud study. Thus, the current work is verified in all kind of atmospheric and environmental conditions around the radar site but only around monsoon seasons of 2013-2016. The typical cases are those (presented in the MS) where the texture differences of reflectivity (with 2-Dim. and HTI plots) and predominant statistical behavior can be clearly seen between insect and cloud. It is evident from that analysis that the biota is confined below 2-2.5 km AGL. For further confirmation of removal of biota, HTI plots for each file in each day have been made automatically within the algorithm for visual re-assurance of the intact cloud vertical structure. Further, presence of biota has also been confirmed using the polarimetric parameters (using earlier published references) from the same radar data, see Figure 11, Figure 13 & 14. We have fixed the maximum low level height as 2.6 km AGL for biota contribution based on reflectivity texture with our manual exposure to all the radar data (i.e., AGL+1.36 km=3.9 km AMSL). In this reference, CBL/ABL depth is not important of

[Figure]

**Figure AR2: TEST performance in filtering biota echoes that are co-present with low clouds.**

**the current idea of the paper and importantly for the hilly, less vegetation radar location.**

AR1-Comment:As an alternative solution to the computationally intensive spectral techniques for the insect clean-up (e.g., Luke et al., 2008), a computationally efficient technique to minimize insect clutter have been implemented based on fuzzy-logic algorithm (e.g., Chandra et al., 2013), which takes into account both the physical properties of clouds and different radar moments. This technique can be applied with different levels of complexity based on the supplemental observations (Microwave Radiometer/Ceilometer) you have in addition to radar moments. I suggest authors go through this technique for more details.

**Response: Yes, We agree about the computationally intensive spectral technique. Hope you may agree with us that spectral technique is memory and labor intensive too!. Importantly, this paper proposes a 'simple' algorithm that makes use of only off-line radar spectral moments profile viz., LDR, Spectral width and Z. Systematic characterization of Z variability using the local atmospheric vertical structure knowledge besides to the theoretical, statistical, and echo tracing tools are the key components of this study.**

AR1-Comment:The basis of the present technique is that the reflectivity distribution could be effective in separating insects from clouds, which may not be the case always. There could be instances when the range of reflectivities from the shallow passive clouds could be similar to the insects (refer to panels, a1 and a2 from the Fig. 13 as in Chandra et al., 2013). This study has taken into account not only the physical properties of cloud (e.g., liquid water path) but also texture signatures in the reflectivity field, the variability of the scatterers inside the radar range resolution from the spectrum width variable-one of the main predictors in insect-cloud separation.

**Response: Yes, we have mainly considered the texture signature with Z. For much clairty, TEST algorithm flowchart Figure 6 at pg 6 and its explanation modified slightly at  pg 7 (point 4). Agree, Our experience with one second radar data is that most of the insects density might be contributed either one or non insect in the radar beam in a second. The above figure AC1 mentioned case has been explained as Figure 13 in MS. (Figure AR2 is complementing to figure 13).**

AR1-Comment:The authors would have shown the technique demonstration effectively with few figures. I feel that there are some figures (Figure 8a and 8b, Figure 11) which don't serve any purpose. Some of the references (cited in the lines 64-98) related to the clutter removing techniques implemented at other frequencies (C- S-Band) were not necessary.

**Response: Yes, optimal usage of Figures has been tried. The purpose of Figure 8a and 8b in this paper is vital since it is inferring the Time-series characteristic difference between smooth meteorological cloud returns with its counterpart, noise or biota. Height time variant natures of noise and biota irregularities (more than 1 dB around mean, Z or its SD) are intermittent whereas such time variability is limited to less than 0.5 around mean Z for cloud. Also it is evident from the Figure 8 that insects de-correlation period is always less**

than 4-5sec.Thus, height time variant nature of Z and corresponding SD gradient is the key for biota identification.  Similarly, Figure 11 demonstrate the important polarimetric capability of the radar as well as to confirm the presence of cloud and biota using polarimetric variables.

---

## Author Comment (AC3) · 24 Nov 2017

We are thankful to the Anonymous Referee 2, for the hearty services in rendering his/her experience and knowledge based comments, those are valuable to us for improving the quality and the focus of the paper. The point-to-point AR2 responses of the authors are as below:

Anonymous Referee #2 (AR2)

AR2-Comment: I think the NER algorithm should be removed from this paper. There are much more general ways for thresholding between signal and "salt and pepper". I should be done by the radar software. so that it adapts automatically to the processing parameters.

[Figure]

Response: NER curves are potential part of the used algorithm required to identify the cloud peak at first place and then backtracked for to its weakly echoing boundary regions. Thus, NER curves are required for complete recovery of cloud structure (see latest figure 2 for point and volume target NER curves). Moreover, the developed algorithm is the part of our automatic off-line data processing software for the quality control of the cloud radar data. And it will also be useful for those who want to use it in the post processing data set.

AR2-Comment: I think the TEST algorithm for filtering insect echoes from the radar data is helpful if it is used in combination with LDR-filtering and or dual frequency filtering. The author comes to this conclusion in the lines around 285 and I agree to it. In the rest of the paper the algorithm is described as a standalone alternative to LDR or dual-frequency filtering. It should be clearly said that this does not work as in regions with much insects the insect signatures are as smooth as butter. There they are volume filling targets.

Response: It has been found with our numerous examples that LDR threshold alone is not able to remove all the biota (e.g., added Figure 13) but inauspiciously affecting the weak cloud portions that are not sufficient enough to excite the cross pol. channel weakest returns due to the cross-pol. isolation restriction of the antenna on the LDR values (see figure 10 and 11 and related discussions at pg 8). Therefore, TEST+LDR filtering is definitely helping for the cases when biota density is more (added Figure 14, pg 37 and its discussion at pg 10) or biota echo co-exists inside the cloud (Figure 13). Still, pure cloud returns are noted to be not possible even by TEST+LDR besides that this combination was also severely affecting the weak cloud portions. Thus, the TEST alone is found to fulfill the requirement significantly of both biota removal as well as recovery of weaker cloud portions using NER curves excepted to cases of high number density of biota or biota existing within the cloud. The NER curves hold the key again here. However, after TEST process, to eliminate further those portions of Z values which are possible biota contamination within cloud inferring from the both LDR

and SW thresholds for the preserving the true cloud returns (see Figure 13i, pg 36).

AR2-Comment: The theoretical background of the algorithm should be explained more general: actually the signal from insects has a longer de-correlation time than signal from volume filling targets.

Response: That could be apparently true if one have high density insect presence with course resolution observations. For our case, insect density is observed to be moderate and that the echo de-correlation time found to be very much shorter than cloud duration. This can be evidently seen with added figure 13, figure 14 and figure A3. Most importantly we demonstrated now with Figure 15 that cloud de-correlate longer than biota those discussion can be seen at page 11 before section 4.

AR2-Comment: The signal from volume targets is a sum of many signals with statistical phases and amplitudes which causes noise with normal distribution (central limit theorem). Therefore even if the volume is filled with stationary targets (droplets falling with different speeds, some exiting the volume, some entering) each line of of the un-averaged complex spectra is normal distributed noise with zero mean and a variance corresponding to the power in the doppler spectrum. The doppler spectra are the abs-square of the complex spectra and therefore they are still noisy. Due to squaring the distribution is transformed from normal to exponential. After averaging over 1 s this noisiness has smoothed out by 1/sqrt(nave). In contrast the signal from a single insect is not noisy at all if its SNR is large. But there is another reason causing variance in the biota signals. Typically the insects are advected through the radar beam, entering with apparent downward velocity and leaving with apparent upward velocity. The pass through time depends on beam width (deg), height, and wind speed. this causes a spiky spectra. if there are not too many insects, then there is a maximum in the variance spectrum of biota signals at 1/(pass through time). For this reason the variance spectrum of volume targets is white and for biota with moderate densities it has a maximum at the frequency corresponding to the 1/(pass through time). The TEST-procedure extracts the variance caused by biota by cancelling the high frequency vari-

**AMTD**

ance of the volume targets by 1 second averaging and by cancelling the low frequency variance with high pass filtering the variance of reflectivities. The remaining medium frequency componets of the variance spectrum is dominated by the beam passing of the single insects, and therefore it can be used for recognising if the signal is from biota or clouds. Without understanding the author found that the test method works in many cases. In cases with too high or too low wind speed this simplified filtering may fail.

Response: The proposed algorithm makes use of time series of 0th moment profile data from the Doppler spectra. So, it is essentially off-line processing of 0th moment time series data for running average of below 5 seconds window. So, there is much concern on biota (insects/birds) number density within the radar sample area than wind speed (observed to be insignificant under light and moderately dense insect condition) as for as TEST performance on off-line moments data is considered. Thus, no need to involve the atmospheric wind or biota velocity details with TEST. To give more clairty further on it, we have chosen two contrasting wind speed day where low level jet (LLJ) shows strong (weak) winds at altitude of $\sim$ 2 km AMSL derived from radar using VAD/VPP method (see belwo figure AR3) and found that TEST filtering working well during both high and low wind speed as well (see below figure AR(4, 5) for performance of TEST).

Some minor notes: AR2-Comment:45: sensible âAËŸT> sensitive

Response: Suggestion has well taken. However it cannot be seen now due to re-writing of Introduction.

AR2-Comment:56: to our experience the reflectivities of biota are below 0 dBZ, reflectivities of rain are above 0 or 5 dBZ.

Response: Okay! Correction made accordingly at line no 70.

AR2-95: T-matrics âAËŸT> Rayleigh

Response: Thank you! Suggestion is implemented at line no 256.

[Figure]

AR2-Comment:96: I would change the sequence from large to small 1 droplets with .1 mm : -60 dBZ 64 droplets with 0.05: -60 dBZ 1e6 droplets with 0.01: -60 dBZ

Response: Thank you! sentence is modified now at line no 259.

AR2-Comment:I guess the author wants to say that hydro meteors are volume filling targets in most cases. For a single spectral component or say a single drop D size Z = N DЁĘ6/V, where V is the radar volume which about 1000 to 25 000 mЁĘ2 depending on height, and N is the number of droplets in the radar volume. In case of single target N=1 and therefore Z_single target = DЁĘ6/V or Z_volume traget Âż DЁĘ6/V. As D can be inferred from the terminal falling velocity which is roughly the doppler velocity at least for larger droplets, it can be found by analysing data that hydrometeors are volume filling in the majority of cases. Sometimes large droplets in the beginning of a rain event are rather single targets.

Response: Yes, we assume that the hydrometeors are mostly volume filling / distributed targets. Agree that single big rain drop case could be point target but that yields very strong reflectivity where identification of cloud is much easy or exclusive in that sense that cloud echo can mask the weaker insect echo.

AR2-Comment:98: is the PRF of this radar really adjusted to such a low value. this would allow for a maximum range of 300 km which is not useful in vertically pointing mode. a prf of 7 to 10 khz is more adequate in vertical mode. this allows a much larger velocity range. but this is not relevant for the scope or this paper.

Response: Yes. Thank you! We used near 5 kHz ie., prt is around 201 micro seconds with maximum range of 30 km. Necessary change made at line 260.

AR2-Comment:I cannot understand or even guess the mening of this sentence. 126: ..more than 2 m/s and the de-correlation

Response: Thanks you for letting us the missed clarity in writing. The mistake has been corrected at line no 290 and such needed clarity and correction can be seen with

subsequent part of the MS. Regarding de-correlation, we are inferring indirectly with our time series echo coherence pertinent to biota and cloud using running average with a hypothesis (see line no 291-295) and subsequently presenting a shallow cumulus cloud presence with biota (figure 13a) case and proving the de-correlation time of biota and cloud echoes using ACF with figure 15.

AR2-Comment:137: This method will be fully explained in the following section âAËŸT> It seems it is in the rest of this section and then in the section Results and discussion beginning in line 214

Response: Agree and implemented at line 321. Thank for the correction suggested.

AR2-Comment:139: fixing âAËŸT> thresholding

Response: Yes, Implemented at line no 323.

AR2-Comment:227: This is not true for cyrus clouds. The have a very soft top.

Response: Hope AR2 means it cirrus clouds, even those clouds have soft top they have to come above the noise floor so it is equally applicable to cirrus clouds as well.

Note: Referee figure quoted as 'Figure ARX' and MS figure as 'Figure X' Please also see pdf responses attached

Please also note the supplement to this comment:
https://www.atmos-meas-tech-discuss.net/amt-2017-254/amt-2017-254-AC3-supplement.pdf

[Figure]

[Figure]

**Fig. 1.** Figure AR3: VAD/VPP based wind profiles from KaSPR volume mode observations on 10 (weak ABL wind) & 24 (strong ABL wind) Sep.2013.

[Figure]

**Fig. 2.** Figure AR4: TEST performance in filtering biota under strong low level Wind Speed (>10 m/s) day at 2.3 km AMSL for three cases on 10 Sep 2013 (i-iii) and case for active monsoon day on 08 Jul 2014 (iv

[Figure]

**Fig. 3.** Figure AR5: TEST performance in filtering biota under weak low level Wind Speed (< 4 m/s) day at 2.3 km AMSL for three cases on 24 Sep 2013 (i-iii) and case for break monsoon day on 28 Jun 2014 (iv).

---

## Author Comment (AC4) · 24 Nov 2017

We are thankful to the Anonymous Referee 3, for the hearty services in rendering his/her experience and knowledge based comments, those are valuable to us for improving the quality and the focus of the paper. The point-to-point AR3 responses of the authors are as below:

Anonymous Referee #3 (AR3)

AR3-Gen. Comment: Millimeter-band radars are very sensitive to detect small targets such as cloud droplets and also insects and other biological particulates (biota) present in great number in the lower atmosphere. Polarization measurement is an efficient mean to discriminate cloud echoes from non meteorological scatterers that

share in common very low reflectivity. Unfortunately most radars are not equipped with polarization measurements. This short paper proposes for these standard radars a simple technique able to separate meteorological and non-meteorological echoes. It uses only successive vertical reflectivity profiles acquired by a 35-GHz radar operated at vertical incidence with a 50 m pulse length and one second temporal sampling. Because of the high spatial and temporal resolution, most of the time only one or no biota target is present in the pulse resolution volume. In contrast, cloud echo is due to millions droplets that fulfill the pulse volume. As a consequence signal variability at a given range between two vertical profiles is much more important for biota scatterers than for cloud echoes. Signal variability is given here by the standard deviation of the reflectivity over the time of five profiles that corresponds to the typical duration of the biota echoes crossing the antenna beam. The threshold value that separates distinctly biota from cloud is obtained from statistical analysis of a large radar observation set. Indeed this value should be adjusted for a radar having different characteristics. The topic of this study enters the scope of the journal and responds to a real issue for anybody who wants to extract physical quantities from radar signal. The work is put into perspective with past equivalent investigations through a large panel of bibliographic references. The work based on well chosen graphics is convincing and above all the methodology is validated with polarization measurements provided by the same radar. In conclusion this paper that presents a good scientific interest is suitable for publication in Atmospheric Measurement Techniques Journal. However this recommendation is subordinated to the authors consideration of the following comments.

Response: we are grateful to the reviewer's learned summary of the work and thankful for intimate resonance with the central idea of the paper. In fact, above concise summary is so fascinated that it has been adopted with little changes at the last section of the Manuscript!. We do agree on the underlined reviewer statement that with little adjustment to TEST, it will be able to work with other radar (please see below figure AR6 where we drop our 1 sec Z measurements of KaSPR (MS figure 7) to every 4 second and 16 second interval

[Figure]

Main review points

AR3-Comment: 1) Lines 48 to 50 give the list of the source of non-meteorological echoes which comprises insects and other biological particulates (biota). The title refers only to insect and in the text the word insect is nearly always used. Even if the insects is the main source of biological echoes it is a restrictive term. I propose to use in place the word biota introduced by the authors.

Response: Agreed and implemented! Insect word replaced with biota for the whole MS.

AR3-Comment: 2)- In figure A2 strong vertical gradient of humidity is associated with the presence of cloud echoes. We may deduce that also strong refractivity index gradient exists which can be a potential source of Bragg or specular echoes. For information an explanation that this type of echoes, observable with UHF and VHF band radars, has a very low probability to be detected by millimeter-band radars will be welcome.

Response: Yes, indeed it is welcome at cloud radar to see clouds at unsaturated elevated cloud layers that is evident at relatively cooler (T $\sim$ -10°C) height level. This may be something that with increasing altitude even relatively less RH close to above 75-80% is sufficient enough to consider as cloud possibly due to the lower saturation vapor pressure associated with predominant ice than water above the zero degree isotherm levels?! ..Speculating! Furthermore! Possible sensitivity of the 35 GHz cloud radar ($\sim$ -36 dBZ; dashed circled region with aside pasted figure AR7 (ref: figure 6 of Kollias et al., BAMS 2007)) to the strongest refractivity index gradient observed to be contributing mainly from huge water vapor gradient (of $\Delta$RH >75% and $\Delta$T < 2°C within $\sim$400 m atmospheric slab centered at $\sim$5.2 km altitude; see Figure A2 of the MS) with Alto-Stratus cloud could have been close correct guess to this happens. This further confirms the sensitivity of the cloud radar to detect weaker shallow depth clouds.

AR3-Comment: 3)- The sensitivity of the radar is -60 dBZ at 1 km range (line 95). This value seems to me very optimistic according to the radar characteristics. Give some

details on the radar calibration.

Response: Yes, KaSPR operated in zenith FFT mode with below configuration: 50 m range resolution, 25 m range gate spacing, 1:10 pulse compression ratio (0.75* 10 i.e., 75% efficient pulse compression), 5 kHz PRF, 128 FFT length/14 coherent averaging, 20 post averaging will have the minimum detectable reflectivity at 1 km is dBZ= -20log(50) − 10log(0.75*10) − 10log(14) − 10log($\sqrt{20}$) +4.4 = -56.3 Where difference between the calibration constant and noise floor (+55.4 − 51 = +4.4) So, the minimum detectable reflectivity at 1 km is -56.3 dBZ (it could be -53.3 dBZ only if a 3dB threshold above Pn/(FFT length*$\sqrt{}$incoherent integration) that yields a false alarm rate of less than 1%).

AR3-Comment: 4)- May be the high radar sensitivity is due to the use of pulse compression (Table 1). If this mode is used give the effective pulse length, the code moments number and the lower range gate available for the data set presented in the paper.

Response: Yes, used the 3.3 $\mu$s pulse length with 10X pulse compression (i.e., compressed to 0.33 $\mu$s in the digital signal processor of the system). So, the radar data set used for this work has the effective pulse length of 50 m and lowest range gate available is at 942 m AGL.

In details, KaSPR employs an improved variation of the well known Linear Frequency Modulated (LFM) pulse compression technique. The KaSPR pulse compression technique is amplitude taper (window) (using a Tukey taper with 0.7 taper coefficient; Window function) on the transmitted LFM pulse and the compression is implemented in the digital signal processor system using a least mean squared filter (Mudukutore et al., 1998) to achieve much improved (lower) range side lobes, compared to un-tapered LFM pulse compressed with a matched filter. Ref: Mudukutore, A., Chandrasekar, V., & Keeler, R. J. (1998). Pulse compression for weather radars. IEEE Transactions on Geoscience and Remote Sensing, 36(1), 125-142. DOI: 10.1109/36.655323. These details are added now in section 2 at pg 3 last para.

AR3-Comment: 5)- The term point target is used line 102 for non-meteorological echo. In fact a scatterer is named punctual echo when it is alone in the pulse volume. In that case echo duration is related to the time taken by the target to cross the radar beam, to its radar cross-section and its position relative to the beam axis. All these factors explain the signal variability of biota echoes.

Response: Yes, agree. In fact the word 'point' used for theos biota target echo that can be seen as point/round discontinuous returns (e.g., figure 2 and figure 13 a). Further with NER curves it has shown they follow point and volume radar equation (see modified figure 2 and below figure AR8).

AR3-Comment: 6)- In fig.1, and others equivalent figures, a range (r) correction of the radar signal of the form r2 is used (line 109). It is correct for volume echoes such as cloud echoes, for point targets it is inadequate. The range correction for such backscatters has the form r4.

Response: Yes, agreed. Suitable modification has been made with the text and figure AR8 as pasted below. The suggested range correction for the possible point target is assumed to be confined mostly below 3 km altitude. These curves are also added now and shown as gray dashed curves with their start point are almost maintained. It is interesting to note that the maximum value of mean noise floor (s14; dashed grey lines) is well within s5 (green) curve that was chosen in this work to first qualify the signal above the noise floor either for cloud or insects echo which has been selected for further process to find the time coherence or correlation periods in the next stage to keep only the cloud. Thus this point has already taken care.

AR3-Comment: 7)- When there is an echo at a certain range, the signal at the receiver output is the sum of the receiver noise voltage and the detected backscattered wave. It is therefore necessary to remove the noise power in order to get the backscattered power. It is evident that this has not been done for the presentation showed in the figures such as fig.1.

Response: Yes, the only spectral moment's profile data has been used in this work (that ensure through signal to noise ratio check for having only backscattered power). This has stated now clealry at the start of section 2. In fact, under weak or no sensible atmospheric targets within the radar sample volume of any radar range gate, the radar spectral moments computation software tracks to pick up the close by background random noise floor peak from the Doppler power spectrum. Actual cloud radar spectra under clear-air condition will have only noise floor at all FFT bins and even at all range gates. It is also quite obvious the case where there was no sensible targets in the cloud radar probing region. Under such void of sensible cloud radar target range gates, the moment's estimation code quite possible to pick up a random noise peak relatively closer to within the Doppler Spectra FFT/velocity bins based on spatial and temporal continuity information. This might have been the reason to have noise Z estimates from the zeroth moment profile. Good thing with this mean background noise is that it is helping to retrieve weaker cloud boundaries some extent using theoretical NER curves.

AR3-Comment: 8)- Line 111: Receiver noise is made of thermal noise generated within the receiver chain and also of other sources which are taken into account through the noise figure of Table 1.

Response: Yes, correction implemented at line no 275-276.

AR3-Comment: 9)- Give more details on the computation of the running mean and standard deviation (line 136) of the successive vertical profiles of reflectivity. In particular it is important to precise if these quantities are computed before or after noise removal.

Response: Mean and standard deviation of the successive vertical profiles of reflectivity (after noise removal) are computed. In fact, we used offline spectral moments data for this entire work. It is first attempted to find out the noise floor using sensitivity (NER) curves and found that S5 curve is near 3-db higher than the maximum observed

noise floor of the KaSPR. Once noise is removed only those echoes are allowed which are higher than S5 curve to segregate cloud and biota. Biota point returns are mostly confined below 3 km altitude with significant shift of mean noise floor just below 1.5 km towards higher (S14 curve; based on the point target NER i.e., r4 X Zstart range) but this still lies well within S5 curve (see above figure at the response of AR3 comment 6) to allow for further process to refine them using standard deviation or time coherence to determine cloud or not. Then de-correlation time of cloud and biota have been found out using running mean and standard deviation of different time interval. Cloud being an meteorological echo changes gradually and so having de correlation period more 40-110 sec. But for insects being spurious in nature it de-correlated quickly, within 4-10 sec. From this computation 4sec has been taken as a key segregator between biota and cloud.

AR3-Comment: 10)- Line 161: Receiver noise is not en echo but a signal generated in the receiver chain.

Response: Admitted the mistake, correction implemented at 1st paragraph of Results and Discussion, line no 342, 361, 362, and 364.

AR3-Comment: 11)- A statistical de-correlation time is introduced line 174. I do not understand very well how it is computed. I think it is related to the standard deviation of the reflectivity. Give the formula that links de-correlation time and reflectivity standard deviation. In figures 3 and in the text the unity used for the standard deviation is not given.

Response: Yes, it is related to the 4-point running mean and standard deviation. (here SD is for Z thus its unit dBZ apply). It is hypothesizing here (provided now in MS at page 290-295) that the running mean and standard deviation of ∼4 seconds reflectivity profiles (i.e., sliding interval of 4 seconds) works in identifying all non-hydrometeor returns. Furthermore, the time coherence of radar returns at every range sample can be checked for every 4 seconds as window period to infer the echo power de-correlation

time or degree of coherence period associated with biota return. In order to prove this, the below figure AR9, is worked out to find the correlation where left panel represents the typical HTI plot of Z measurements for low level/ shallow cumulus cloud in the presence of biota and right panel shows the simple auto correlation function (ACF) having lag (0-300 sec) correlation corresponding to the reflectivity time series of shallow cumulus cloud (base, mid, top) and biota heights at 1.5 and 2.6 km. From the ACF analysis it is clear that biota shows quicker ($\sim$4 seconds) de-correlations periods than cloud ($\sim$ 40-170 seconds). It is also to be noted that clouds may show varied de-correlation periods above 30 seconds but insects mostly de-correlate very much less than 10 seconds. Hence, the hypothsis for TEST proves here with. These discussions and newly added figures (13-15) are can be seen with MS at page 4, 10-11 and 37-39 .

AR3-Comment: 12)- lines 218 to 219 ...biota that are found to extend less than 2-4 height bins each of 25 m... : vertical spreading of a point echo is expected to extend over half pulse. How do you explain this large spreading that can approach 2 pulse lengths. Is the use of a compressed pulse that produces this increase.

Response: Yes partially. In fact, the used pulse width is 3.33 $\mu$s with 10X LFM chirp compression with sampling in range (range gate spacing) at every 25 m. So, the uncompressed range bin width of $\sim$500 m that become 50 m after 10X pulse compression. It is quite possible that biota movement can confine sometime in-between two range gates then the biota echo spreading can confine maximum of 100 m. This could be the reason. However, small correction has now made in the MS that biota echo extends maximum of 2 range gate intervals of each 50 m or 4 range gate spacing of each 25 m. See these detials with MS at line no. 368-370, pg 6, and line no. 438-439 page 7 and modified first Para of MS Section 2.

AR3-Comment: 13)- I suppose that the radar has Doppler capability because line 263 and 264 PulsePair and Fourier Transform are cited. Doppler spectra width contains information at the pulse scale on the de-correlation time of the echoes. It could have

been used instead of the reflectivity standard deviation. Did you try to analyze this quantity to discriminate echo type.

Response: Yes, KaSPR having Doppler capability and the 2nd moment velocity variance/Spectral width measurements are available. Thanks to referee that TEST results has now been able to cross checked and found that less spectral width values (∼0.3 m2s-2) confirmed the shorter coherence time / short temporal correlation associated with biota. Thus TEST, used running mean and S.D from set of 4 profiles, is working to ensure the biota and cloud through their de-correlation time less than 5 sec. interval. Therefore, TEST is simple but potential because that makes use of single Z parameter but critically through to track its change both at spatial and temporal levels. However, TEST output Z needs to further constrained with SW and LDR thresholds that are found be advantageous to have best possible cloud only radar returns mainly within cloud region. New Figure 13 and the relevant discussions have been added in this regard to the MS at page 10-11.

Note: Referee figure quoted as 'Figure ARX' and MS figure as 'Figure X' Please also see pdf responses attached

Please also note the supplement to this comment:
https://www.atmos-meas-tech-discuss.net/amt-2017-254/amt-2017-254-AC4-supplement.pdf

———————————————

[Figure]

**Fig. 1.** Figure AR6: KaSPR 1s (see figure 7 in MS) resolution Z profiles are re-sampled at 4s (left three panels) and 16 S. Biota echo seen differently at different time interval sampling.

[Figure]

**Fig. 2.** Figure AR7: Atmospheric radar echo scattering Vs radar wavelengths (taken from Kollias et al., BAMS 2007).

[Figure]

**Fig. 3.** Figure AR8: Radar Sensitivity curves are now using range correction to the radar backscattering based on the volume (r^2 form) and point radar equation (r^4 form).

[Figure]

**Fig. 4.** Figure AR9: Shallow cumulus cloud present with biota (HTI plot) and (right panel) is AFC based 0-300 lag correlation for cloud and biota.

---

## Author Response (AR1)

**Dear Dr Gianfranco Vulpiani, AMT Handling Editor, EGU-Copernicus**

We are thankful to you for your quick decision on our initial submitted work 'publishes as is' on 18 Aug 2017. We have been critically worked out and coming now with extensive revision by accommodating all comments of the AMTD three reviewers. The point-to-point comment and responses with the detailed implementation are prepared separately for the three referees that are followed after this cover letter. The track-change word file of the MS that reflect all the review response actions is also prepared.

We happy that through this revision, our MS mainly improved on

- Proposed TEST algorithm performance under cloud with biota presence (figure 13-14) is answered thoroughly (thanks to AR1 concern). Further, using LDR (thanks to AR2) and SW (thanks to AR3), TEST is now able to work under high number concentrated biota and within cloud biota cases which was actually the weakness of TEST earlier.
- Inferring biota and cloud De-correlation periods (thanks to AR3 and AR1) are now supported (figure 15)
- More relevant technical details (AR3) and pertinent references (AR1) are provided.
- Potential of the current work is highlighted now (AR1).

In fact, we are grateful to you, the Editor(s) and all Editorial team for their services/help and untiring timely support and cooperation. We are also equally thankful to all the three Anonymous Referees, for their hearty services in rendering experience and knowledge based comments, those were valuable in improving the quality and the focus of the paper.

Thanks-in-advance.

Sincerely Yours, Madhu Chandra Reddy Kalapureddy (email: kalapureddy1@gmail.com)

**PS:**

1. This covering letter include Author responses to comments at Page 2-7 for AR1, Page 8-14 for AR2 and Page 9-22 for AR3.

2. modified Manuscript with figures

: AMT2017b25-MS pdf extension

**1** | Page

**Authors Responses**

To the Interactive comments on manuscript titled "Simple insect removal algorithm for 35- GHz cloud radar measurements", M C R Kalapureddy et al. Atmos. Meas. Tech. Discuss., doi:10.5194/amt-2017-254, 2017

At the outset, we are grateful to the Editor(s) and all Editorial team for their services/help and untiring timely support and cooperation. We are also equally thankful to all the three Anonymous Referees, for their hearty services in rendering experience and knowledge based comments, those are valuable to us for improving the quality and the focus of the paper.

The point-to-point AR1 responses of the authors are as below:

**Anonymous Referee #1 (AR1)**

AR1-Comment: The study presented a technique which uses high temporal and spatially resolved reflectivity profiles to extract the cloud echoes from the clutter (mainly from the biota). The proposed technique suggested as a simple and efficient solution for clutter removal, compared to earlier sophisticated techniques based on dual polarization and spectral techniques. I think manuscript has several shortcomings, related to technique and assumptions, poor job of literature review, references and lack of solid conclusions. In its entirety, I would recommend rejection of this paper in its present form.

Response: Thank you! We request AR1 now to review the latest modified version of the paper where we re-written the whole introduction part the manuscript (MS) and cited all possible concern references that come under the scope of the paper and responded also to other valuable referee points. Furthermore, clarity on the technique/algorithm and its main application region has now been clearly come through the revision process in the current modified version of the MS at first Para of section 2 (pg 3) and added basis for TEST (Line 292-295, pg 4-5), including new figures (fig 13 to fig 15) show the potential of TEST in screening out clouds by filtering out biota. Further weakness of TEST under challenging conditions like within cloud and high density biota has been overcome using extra measurements like LDR and SW. This can be seen at the last two paragraphs of Results and Discussion. Thus, the main conclusion of the paper is how simplest way one can remove the biota contribution and preserve true cloud hydrometeor echo and its need for the study of important shallow cumulus/ABL clouds before the actual cloud radar echo weighted measurements consider for any research application purpose. The above revision asked necessary modification to the last section (Summary and Conclusions) from page 11 onwards.

AR1-Major comments: The screening technique authors have implemented using simple measures of reflectivity (or SNR) thresholds and its variability to filter out the clutter has been a usual practice in the cloud radar community as a part of post-processing exercise. The challenge of separating insects from the cloud clutter is difficult due to the lack of clear demarcation between their properties as seen by cloud radar. More often than otherwise, the screening process requires more than one variable, which captures the texture, distribution width, and physical properties of these echoes. With this motivation, some of the earlier studies have devoted their efforts to address this problem using different techniques (fuzzy-logic, spectral technique or polarization properties).

Response: Separating biota from cloud is difficult and challenging but not impossible by the cloud radar if one effectively makes use of advancing radar and signal processing technique (e.g., chirping and DSP) that enables to have the provision of high spatial and temporal resolution radar measurements (1st paragraph of System, Data and Methodology) that can demarcate the cloud echo from insects for example through reflectivity texture (e.g., TEST). With our knowledge, TEST, it is first of its kind effort to consider both reflectivity variance (i.e., dBZ texture) and its rate of change through running average for every 4 seconds. The above point pragmatically working as to identify the time coherence or de-correlation periods associated with clouds and biota echo signature (see newly added figure 15 and its description at pg 11). Moreover, the de-correlation can be evidenced through direct third Doppler power spectral moment; spectral width measurements that clearly show biota exhibits less velocity variance thus the relatively quicker time decorrelation at the pulse scale. In Fig. 2, the zoomed portion of Fig 1, the rounded echo confirms the presence of non-hydrometeor information by their duration of maximum 10 sec which is too small for a cloud to form and then suddenly disappear. So the vertical extension and the time duration of the echoes are the two key factors to discriminate cloud from non-meteorological information. Merits and de-merits of TEST has been brought out exclusively with Figure 13 and 14 that are making use of LDR and Spectral width measurements besides to Z to enhance the proposed TEST algorithm capability under tough conditions like cloud under heavily dense insects clutter.

AR1-Comment: The authors haven't clearly appreciated and addressed the insect removal to the detail that it was needed. They have demonstrated the algorithm with several minutes of data, which doesn't warrant the techniques robustness to apply for other conditions. Authors have made several assumptions about the insect layer depth, their decorrelation timescale without presenting any evidence about the location of the shallow boundary layer clouds, where the insect clutter is very critical. Previous studies (e.g., Geerts and Miao 2005; Chandra et al., 2010) have utilized the long-term observations of insect echoes to study the convective boundary layer, where they have shown that the insect decorrelation times may vary from few seconds to few minutes depends on boundary layer organization. The authors would have shown the distribution of the cloud base locations (from the closest ceilometer data) to justify their presumed insect layers below  $\sim$ 2km. I suggest authors to utilize the supplemental observations (such as

ceilometer, microwave radiometer) to present the cloud properties and refine their insect-cloud algorithm based on the locations of cloud layer depth.

Response: Unfortunately the suggested useful complemented data was not available at and around the radar site. However using some available GPS RS observations from the radar site, the presence of weaker clouds have been proved with auxiliary Figure A2 (please also see the response to the comment 2 of AR#3). Further, we consider the reviewer's well suggested point on the inclusion of shallow boundary layer cloud case with insects clutter (Figure AR1 and AR2) when both have near same reflectivity values (added Figures 13-14). In fact, thanks to the reviewer that now it is clearly illustrating the potential of TEST that lies mostly to the ABL, where shallow cloud evolves, where the affinity of biota are predominant. Below are two examples of such low level/ shallow cumulus clouds with biota clutter where the fine performance of TEST is evident.

---

## Referee Report (RR1)

**Review for:** Atmospheric Measurement Techniques
**Manuscript number:** AMT 2017-254

**Title:** A *simple biota removal algorithm for 35-GHz cloud radar  measurements.*

**Authors:**Madhu Chandra R. Kalapureddy, Sukanya Patra, Subrata K Das, Sachin M Deshpande, Kaustav Chakravarty1, Ambuj K Jha1, Prasad Kalekar, Hari Krishna Devisetty, Andrew L Pazamany and Pandithurai Govindan.

**Report on the revised manuscript and recommendation for disposition of the paper**

 Author's responses to the questions I raised in the previous review are satisfactory. They followed quite closely my comments. they have made a real effort to improve the clarity and the technical and scientific content of the text following the comments of the different referees. New or corrected figures are added and new analyses or developments and explanations are presented and discussed. The arguments given in response to the referees criticisms are on the whole convincing. The main reproach which concerns the ability of the presented algorithm to work in all situation is certainly justified but the same comment can be made for previous more sophisticated equivalent techniques. Comparison with polarization measurements provides results from which future readers can form an opinion on the efficiency of the technique.
    Better and simpler technique for discrimination between insects echoes and pure atmospheric echoes will be useful for both meteorologists and entomologists but indeed with opposite interest. The use of radar by entomologists to investigate insect behavior in the atmosphere is a growing research field. It is now well established that insect population is strongly decreasing. Observations made by entomologic radars might help to understand this dramatic evolution. It is one of the reasons to encourage scientific groups involved in radar data analysis. I consider that the paper under review is an honest work and as a consequence I recommend the publication of the revised manuscript in Atmospheric Measurement Techniques Journal.